# FINE-TUNING OF CONTINUOUS-TIME DIFFUSION MODELS AS ENTROPY-REGULARIZED CONTROL

## ABSTRACT

Diffusion models excel at capturing complex data distributions, such as those of natural images and proteins. While diffusion models are trained to represent the distribution in the training dataset, we often are more concerned with other properties, such as the aesthetic quality of the generated images or the functional properties of generated proteins. Diffusion models can be finetuned in a goal-directed way by maximizing the value of some reward function (e.g., the aesthetic quality of an image). However, this may lead to reduced sample diversity, significant deviations from the training data distribution, and even poor sample quality due to the exploitation of an imperfect reward function. The last issue often occurs when the reward function is a learned model meant to approximate a ground-truth "genuine" reward, as is the case in many practical applications (e.g., using a learned estimator of aesthetic quality). These challenges, collectively termed "overoptimization," pose a substantial obstacle. To address this overoptimization, we frame the fine-tuning problem as entropy-regularized control against the pretrained diffusion model, i.e., directly optimizing entropy-enhanced rewards with neural SDEs. We present theoretical and empirical evidence that demonstrates our framework is capable of efficiently generating samples with high genuine rewards, mitigating the overoptimization of imperfect reward models.

## 1 INTRODUCTION

Diffusion models have gained widespread adoption as effective tools for modeling complex distributions (Sohl-Dickstein et al., 2015; Song et al., 2020; Ho et al., 2020). These models have demonstrated state-of-the-art performance in various domains such as image generation and biological sequence generation (Jing et al., 2022; Wu et al., 2022). While diffusion models effectively capture complex data distributions, our primary goal frequently involves acquiring a finely tuned sampler customized for a specific task using the pre-trained diffusion model as a foundation. For instance, in image generation, we might like to fine-tune diffusion models to enhance aesthetic quality. In biology, we might aim to improve bioactivity. Recent endeavors have pursued this objective through reinforcement learning (RL) (Fan et al., 2023; Black et al., 2023) as well as direct backpropagation through differentiable reward functions (Clark et al., 2023; Prabhudesai et al., 2023). Such reward functions are typically learned models meant to approximate a ground-truth "genuine" reward; e.g., an aesthetic classifier is meant to approximate the true aesthetic preferences of human raters.

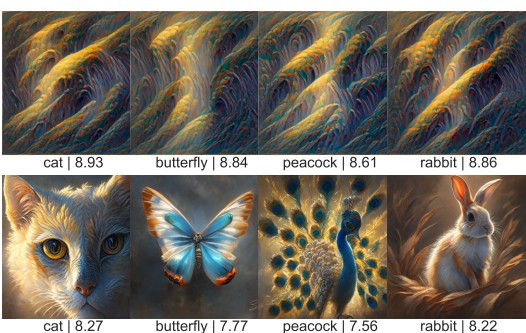

Figure 1: **Mitigating overoptimization with entropy-regularized control.** Diffusion models fine-tuned in a goal-directed manner can produce images (top) with high nominal reward values such as aesthetic scores. However, these images lack realism because the naïve fine-tuning process is not incentivized to stay close to the pre-trained data distribution. Our approach (bottom) mitigates this issue via entropy-regularized stochastic optimal control.

While these methods allow us to generate samples with high "nominal" (approximate) rewards, they often suffer from *overoptimization (reward collapse)*. Overoptimization manifests as fine-tuned models produce samples with low genuine rewards that are still scored as having a high "nominal" reward under the (learned) reward model, as illustrated in Figure 1. This issue arises because nominal rewards are usually learned from a finite training set to approximate the genuine reward function, meaning that they are accurate only within their training distribution. Consequently, fine-tuning methods quickly exploit nominal rewards by moving beyond the support of this distribution.

Our goal in this paper is to develop a principled algorithmic framework and its fundamental theory for fine-tuning diffusion models that both optimize a reward function and stay close to the training data, thus alleviating overoptimization. To achieve this, we frame the fine-tuning of diffusion models as an entropy-regularized control problem. It is known that diffusion models can be formulated as stochastic differential equations (SDEs) with a drift term and a diffusion term (Song et al., 2020). Based on this formulation, in a fine-tuning step, we consider solving stochastic control by neural SDEs in a computationally efficient manner. Here, we introduce a loss that combines a terminal reward with entropy regularization against the pre-trained diffusion model and optimize with respect to both a drift term and an initial distribution. This entropy-regularization term enables us to maintain the bridges (i.e., the posterior distributions of trajectories conditioned on a terminal point) of pre-trained diffusion models, akin to bridge-matching generative models (Shi et al., 2023), such that the fine-tuned diffusion model avoids deviating too much from the pre-trained diffusion model.

Notably, we theoretically show that the fine-tuned SDE, optimized for both the drift term and initial distribution, can produce specific distributions with high nominal rewards that are within the support of their training data distribution. Hence, our approach effectively mitigates the overoptimization problem since nominal rewards accurately approximate genuine rewards in that region. Furthermore, our theoretical results shed light on an intriguing new connection with classifier guidance (Dhariwal and Nichol, 2021).

Our contribution can be summarized as follows: we introduce a computationally efficient, theoretically and empirically supported method for fine-tuning diffusion models: **ELEGANT** (finE-tuning doubLe Entropy reGulArized coNTrol) that excels at generating samples with high genuine rewards. While existing techniques in image generation (Fan et al., 2023; Prabhudesai et al., 2023; Clark et al., 2023) include components for mitigating overoptimization, we demonstrate stronger theoretical support by explicitly characterizing target distributions in our key Theorem 1 (among methods that directly backpropagate through differentiable rewards) and superior empirical performance (compared to a KL-penalized PPO). Additionally, unlike prior work, we apply our method to both image generation and biological sequence generation, demonstrating its effectiveness across multiple domains.

## 2 RELATED WORKS

We provide an overview of related works. We leave the discussion of our work, including fine-tuning LLMs, sampling with control methods, and MCMC methods to Appendix A.

**Diffusion models.** Denoising diffusion probabilistic models (DDPMs) create a dynamic stochastic transport using SDEs, where the drift aligns with a specific score function (Song et al., 2020; Ho et al., 2020). The impressive performance of DDPMs has spurred the recent advancements in bridge (flow)-matching techniques, which construct stochastic transport through SDEs with drift terms aligned to specific bridge functions (Liu et al., 2022; Shi et al., 2023; Tong et al., 2023; Lipman et al., 2023; Somnath et al., 2023; Liu et al., 2023; Delbracio and Milanfar, 2023; Shi et al., 2023).

**Guidance.** Dhariwal and Nichol (2021) introduced classifier-based guidance, an inference-time technique for steering diffusion samples towards a particular class. More generally, guidance uses an auxiliary differentiable objective (e.g., a neural network) to steer diffusion samples towards a desired property (Graikos et al., 2022; Bansal et al., 2023). In our experiments, we show that our fine-tuning technique outperforms a guidance baseline that uses the gradients of the reward model to steer the pre-trained diffusion model toward high-reward regions.

**Fine-tuning as RL/control.** Lee et al. (2023); Wu et al. (2023) employ supervised learning techniques to optimize reward functions, while Black et al. (2023); Fan et al. (2023) employ an RL-based method to achieve a similar goal. Clark et al. (2023); Xu et al. (2023); Prabhudesai et al. (2023) present a fine-tuning method that involves direct backpropagation regarding rewards, which bears some resemblance to our work. Nevertheless, there are several notable distinctions between

our approaches. Specifically, we incorporate an entropy-regularization term and also learn an initial distribution, both of which play a critical role in targeting the desired distribution. We present novel theoretical results that demonstrate the benefits of our approach, and we provide empirical evidence that our method more efficiently mitigates reward collapse.

It is worthwhile to note that while Fan et al. (2023) incorporates KL regularization, there are notable differences in several aspects.

- The training algorithms employed are fundamentally distinct, as our approach is control-based, whereas their training algorithm relies on PPO. Hence, our optimization algorithm can directly control the KL term compared to PPO-based optimization. This enables us to minimize the KL term more effectively while maintaining higher reward values. Consequently, our method better mitigates overoptimization, as we will empirically demonstrate in Section 8.2.

- The PPO-based algorithm is computationally slower, as we will show in Section 8.3.

- We provide theoretical support by explicitly deriving our target distribution in Theorem 1. Based on this, we argue that the fine-tuning algorithm mitigates overoptimization from a statistical perspective, as the fine-tuned distribution retains the same support as the pre-trained distribution, as shown in Section 4. Since detecting overoptimization in real experiments is challenging due to the often unknown true rewards, we believe having this theoretical guarantee is a significant advantage. Lastly, it is worthwhile to note our result highlights a non-trivial connection with classifier guidance, as we show in Section 5.2.

## 3 PRELIMINARIES

We briefly review current continuous-time diffusion models. A diffusion model is described by the following SDE:

$$dx_t = f(t, x_t)dt + \sigma(t)dw_t, \quad x_0 \sim \nu_{\text{ini}} \in \Delta(\mathbb{R}^d), \tag{1}$$

where $f : [0, T] \times \mathbb{R}^d \to \mathbb{R}^d$ is a drift coefficient, and $\sigma : [0, T] \to \mathbb{R}_{>0}$ is a diffusion coefficient associated with a $d$-dimensional Brownian motion $w_t$, and $\nu_{\text{ini}}$ is an initial distribution such as a Gaussian distribution. Note that many papers use the opposite convention, with $t = T$ corresponding to the initial distribution and $t = 0$ corresponding to the data. When training diffusion models, the goal is to learn $f(t, x_t)$ from the data at hand so that the generated distribution from the SDE (1) corresponds to the data distribution through score matching (Song et al., 2020) or bridge/flow matching (Liu et al., 2022). For details, refer to Appendix D.

In our work, we focus on cases where we have such a pre-trained diffusion model (i.e., a pre-trained SDE). Denoting the density at time $T$ induced by the pre-trained SDE in (1) as $p_{\text{data}} \in \Delta(\mathbb{R}^d)$, this $p_{\text{data}}$ captures the intricate structure of the data distribution. In image generation, $p_{\text{data}}$ captures the structure of natural images, while in biological sequence generation, it captures the biological space. **Notation.** We often consider a measure $\mathbb{P}$ induced by an SDE on $\mathcal{C} := C([0, T], \mathbb{R}^d)$ where $C([0, T], \mathbb{R}^d)$ is the whole set of continuous functions mapping from $[0, T]$ to $\mathbb{R}^d$ (Karatzas and Shreve, 2012). The notation $\mathbb{E}_{\mathbb{P}}[f(x_{0:T})]$ means that the expectation is taken for $f(\cdot)$ w.r.t. $\mathbb{P}$. We denote $\mathbb{P}_t$ as the marginal distribution over $\mathbb{R}^d$ at time $t$, $\mathbb{P}_{s,t}(x_s, x_t)$ the joint distribution over $\mathbb{R}^d$ time $s$ and $t$, and $\mathbb{P}_{s|t}(x_s|x_t)$ the conditional distribution at time $s$ given time $t$. We also denote the distribution of the process pinned down at an initial and terminal point $x_0, x_T$ by $\mathbb{P}_{\cdot|0,T}(\cdot|x_0, x_T)$ (we similarly define $\mathbb{P}_{\cdot|T}(\cdot|x_T)$). With a slight abuse of notation, we exchangeably use distributions and densities [1] We defer all proofs to Appendix C.

## 4 DESIRED PROPERTIES FOR FINE-TUNING

In this section, we elucidate the desired properties for methods that fine-tune diffusion models. With a reward function $r : \mathbb{R}^d \to \mathbb{R}$, such as aesthetic quality in image generation or bioactivity in biological sequence generation, our aim is to fine-tune a pre-trained diffusion model so as to maximize this reward function, for example to generate images that are more aesthetically pleasing.

However, the "genuine" reward function (e.g., a true human rating of aesthetic appearance) is usually unknown, and instead a computational proxy must be learned from data — typically from the same

---

[1] We sometimes denote densities such as $d\mathbb{P}_T/d\mu$ by just $\mathbb{P}_T$ where $\mu$ is Lebesgue measure.

or a similar distribution as the pre-training data for the diffusion model. As a result, while $r(x)$ may be close to the genuine reward function within the support of $p_{\text{data}}$, it might not be accurate outside of this domain. More formally, by denoting the genuine reward by $r^\star$, a nominal reward $r$ is typically learned as

$$r = \text{argmin}_{r' \in \mathcal{F}} \sum_i \{r^\star(x^{(i)}) - r'(x^{(i)})\}^2],$$

where $\{x^{(i)}, r^\star(x^{(i)})\}_{i=1}^n$ is a dataset, and $\mathcal{F}$ is a function class (e.g., neural networks) mapping from $\mathbb{R}^d$ to $\mathbb{R}$. Under mild conditions, it has been shown that in high probability, the mean square error on $p_{\text{data}}$ is small, i.e.,

$$\mathbb{E}_{x \sim p_{\text{data}}}[\{r^\star(x) - r(x)\}^2] = O(\sqrt{\text{Cap}(\mathcal{F})/n}),$$

where $\text{Cap}(\mathcal{F})$ is a capacity of $\mathcal{F}$ (Wainwright, 2019). However, this does not hold outside of the support of $p_{\text{data}}$.

Taking this into account, we aim to fine-tune a diffusion model in a way that preserves three properties: (a1) the ability to generate samples with high rewards, and (a2) ensuring sufficient proximity to the initial pre-trained diffusion (1). In particular, (a2) helps avoid overoptimization because learned reward functions tend to be accurate on the support of $p_{\text{data}}$.

To accomplish this, we consider the optimization problem:

$$p_{\text{tar}} = \underset{p \in \Delta(\mathbb{R}^d)}{\text{argmax}} \underbrace{\mathbb{E}_{x \sim p}[r(x)]}_{\Psi(1)} - \alpha \underbrace{\text{KL}(p \| p_{\text{data}})}_{\Psi(2)}, \tag{2}$$

where $\alpha \in \mathbb{R}_{>0}$ is a hyperparameter. The initial reward term $\Psi(1)$ is intended to uphold the property (a1), while the second entropy term $\Psi(2)$ is aimed at preserving the property (a2).

It can be shown that the target distribution in (2) takes the following analytical form:

$$p_{\text{tar}}(x) = \exp(r(x)/\alpha))p_{\text{data}}(x)/C_{\text{tar}}, \tag{3}$$

where $C_{\text{tar}}$ is a normalizing constant. Therefore, the aim of our method is to provide a tractable and theoretically principled way to emulate $p_{\text{tar}}$ as a fine-tuning step.

### 4.1 IMPORTANCE OF KL REGULARIZATION

Before explaining our approach to sample from $p_{\text{tar}}$, we elucidate the necessity of incorporating entropy regularization term in (2). This can be seen by examining the limit cases as $\alpha$ tends towards $0$ and when we fix $\alpha = 0$ a priori. To be more precise, as $\alpha$ approaches zero, $p_{\text{tar}}$ tends to converge to a Dirac delta distribution at $x_{\text{tar}}^\star$, defined by: $x_{\text{tar}}^\star = \text{argmax}_{x \in \mathbb{R}^d : p_{\text{data}}(x) > 0} r(x)$. This $x_{\text{tar}}^\star$ represents an optimal $x$ within the support of $p_{\text{data}}$. Conversely, if we directly solve (2) with $\alpha = 0$, we may venture beyond the support: $x^\star = \text{argmax}_{x \in \mathbb{R}^d} r(x)$. This implies that the generated samples might no longer adhere to the characteristics of natural images in image generation or biological sequences within the biological space. As we mentioned, since $r(x)$ is typically a learned reward function from the data, it won't be accurate outside of the support of $p_{\text{data}}(x)$. Hence, $x^\star$ would not have a high genuine reward, which results in "overoptimization". For example, this approach results in the unnatural but high nominal reward images in Figure 1.

## 5 ENTROPY-REGULARIZED CONTROL WITH PRE-TRAINED MODELS

We show how to sample from the target distribution $p_{\text{tar}}$ using entropy-regularized control.

### 5.1 STOCHASTIC CONTROL FORMULATION

To fine-tune diffusion models, we consider the following SDE by adding an additional drift term $u$ and changing the initial distribution of (1):

$$dx_t = \{f(t, x_t) + u(t, x_t)\}dt + \sigma(t)dw_t, x_0 \sim \nu, \tag{4}$$

where $u(\cdot, \cdot) : [0, T] \times \mathbb{R}^d \to \mathbb{R}$ is a drift coefficient we want to learn and $\nu \in \Delta(\mathbb{R}^d)$ is an initial distribution we want to learn. When $u = 0$ and $\nu = \nu_{\text{ini}}$, this reduces to a pre-trained SDE in (1). Our objective is to select $u$ and $\nu$ in such a way that the density at time $T$, induced by this SDE, corresponds to $p_{\text{tar}}$.

Now, let's turn our attention to the objective function designed to achieve this objective. Being motivated by (2), the objective function we consider is as follows:

$$u^\star, \nu^\star = \underset{u,\nu}{\operatorname{argmax}} \, \mathbb{E}_{\mathbb{P}^{u,\nu}} \underbrace{[r(x_T)]}_{(b1)} - \frac{\alpha}{2} \underbrace{\mathbb{E}_{\mathbb{P}^{u,\nu}} \left[ \int_{t=0}^T \frac{\|u(t,x_t)\|^2}{\sigma^2(t)} dt + \log\left(\frac{\nu(x_0)}{\nu_{\mathrm{ini}}(x_0)}\right) \right]}_{(b2)}, \quad (5)$$

where $\mathbb{P}^{u,\nu}$ is a measure over $\mathcal{C}$ induced by the SDE (4) associated with $(u,\nu)$. Within this equation, component (b1) is introduced to obtain samples with high rewards. This is equal to $\Psi(1)$ in (2) when $p(\cdot)$ in (2) comes from $\mathbb{P}_T^{u,\nu}$. The component (b2) corresponds to the KL divergence over trajectories: $\mathrm{KL}(\mathbb{P}^{u,\nu}(\cdot)\|\mathbb{P}^{\mathrm{data}}(\cdot))$ where $\mathbb{P}^{\mathrm{data}}$ is a measure over $\mathcal{C}$ induced by the pre-trained SDE (1), which has been proved by using Girsanov theorem. In particular, this is actually equal to $\Psi(2)$ in (2) under optimal control, as we will see soon in the proof of our key theorem.

We can derive an explicit expression for the marginal distribution at time $t$ under the distribution over $\mathcal{C}$ induced by the SDE associated with the optimal drift and initial distribution denoted by $\mathbb{P}^\star$ (i.e., $\mathbb{P}^{u^\star,\nu^\star}$). Here, we define the optimal (entropy-regularized) value function as

$$V_t^\star(x) = \mathbb{E}_{\mathbb{P}^\star} \left[ r(x_T) - \frac{\alpha}{2} \int_{k=t}^T \frac{\|u^\star(k,x_k)\|^2}{\sigma^2(k)} dk \Big| x_t = x \right].$$

**Theorem 1** (Induced marginal distribution). *The marginal density at step $t \in [0,T]$ under the diffusion model with a drift term $u^\star$ and an optimal initial distribution $\nu^\star$ (i.e., $\mathbb{P}_t^\star$) is*

$$\mathbb{P}_t^\star(\cdot) = \exp(V_t^\star(\cdot)/\alpha)\mathbb{P}_t^{\mathrm{data}}(\cdot)/C_{\mathrm{tar}}$$

*where $\mathbb{P}_t^{\mathrm{data}}(\cdot)$ is a marginal distribution at $t$ of $\mathbb{P}^{\mathrm{data}}$ over $\mathcal{C}$.*

This marginal density comprises two components: the optimal value function term and the density at time $t$ induced by the pre-trained diffusion model. Note that the normalizing constant $C_{\mathrm{tar}}$ is independent of $t$.

Crucially, as a corollary, we observe that by generating a sample following the SDE (4) with $(u^\star, \nu^\star)$, we can sample from the target $p_{\mathrm{tar}}$ at the final time step $T$. Furthermore, we can also determine the explicit form of $\nu^\star$.

**Corollary 1** (Justification of control problem). $\mathbb{P}_T(\cdot) = p_{\mathrm{tar}}(\cdot)$.

**Corollary 2** (Optimal initial distribution). $\nu^\star(\cdot) = \exp(V_0^\star(\cdot)/\alpha)\nu_{\mathrm{ini}}(\cdot)/C_{\mathrm{tar}}$.

In the following section, to gain deeper insights, we explore two interpretations.

### 5.2 Feynman–Kac Formulation

We see an interpretable formulation of the optimal value function. Importantly, we use this form to learn the optimal initial distribution later in our algorithm (Algorithm 1 ) and the proof of Theorem 1. Furthermore, this result highlights a non-trivial connection with classifier guidance.

**Lemma 1** (Feynman–Kac Formulation). $\exp\left(\frac{V_t^\star(x)}{\alpha}\right) = \mathbb{E}_{\mathbb{P}^{\mathrm{data}}} \left[ \exp\left(\frac{r(x_T)}{\alpha}\right) | x_t = x \right]$.

This lemma has been mainly proved by Feynman–Kac formula (Shreve et al., 2004). It illustrates that the value function at $(t,x)$ is higher when it allows us to hit regions with high rewards at $t = T$ by following the pre-trained diffusion model afterward. Invoking the Hamilton–Jacobi–Bellman equation and using this optimal value function, we can write the optimal drift $u^\star(t,x)$ as $\sigma^2(t)\nabla_x V_t^\star(x)/\alpha$. By plugging Lemma 1 into $\sigma^2(t)\nabla_x V_t^\star(x)/\alpha$, we obtain the following.

**Lemma 2** (Optimal drift). $u^\star(t,x) = \sigma^2(t)\nabla_x \left\{ \log \mathbb{E}_{\mathbb{P}^{\mathrm{data}}} \left[ \exp\left(\frac{r(x_T)}{\alpha}\right) | x_t = x \right] \right\}$.

It says the optimal control aims to move the current state $x$ at time $t$ toward a point where it becomes easier to achieve higher rewards after following the pre-trained diffusion.

**Connection with Classifier Guidance.** The theoretical result Lemma 2 is notable because it simplifies to the formulation used in classifier guidance when rewards are set as classifiers. Specifically, by defining $r$ as $p(y|x) : \mathcal{X} \to \Delta(\mathcal{Y})$, where $\mathcal{Y}$ is a class label and $\alpha = 1$, the optimal drift reduces to:

$$u^\star(t, x) = \sigma^2(t)\nabla_x \log p(y|x_t = x), \quad p(y|x_t) = \mathbb{E}_{\mathbb{P}^{\text{data}}}[p(y \mid x_T)|x_t].$$

This is the well-known form of classifier guidance (Dhariwal and Nichol, 2021).

The above suggests that classifier guidance is mathematically targeting the same distribution as our approach. To our knowledge, this interesting connection has not yet been recognized in the existing literature. However, empirically, the performance can differ significantly due to function approximation and optimization errors. More specifically, in our algorithm, we don't need to explicitly estimate value functions, unlike classifier guidance. We will present the empirical comparison in Section 8.

### 5.3 BRIDGE PRESERVING PROPERTY

We start by exploring more explicit representations of joint and conditional distributions to deepen the understanding of our control problem.

**Lemma 3** (Joint distributions). *Let $0 \le s < t \le T$. Then,*

$$\mathbb{P}^\star_{s,t}(x, y) = \mathbb{P}^{\text{data}}_{s,t}(x, y) \exp(V^\star_t(y)/\alpha)/C_{\text{tar}}, \mathbb{P}^\star_{s|t}(x|y) = \mathbb{P}^{\text{data}}_{s|t}(x|y). \tag{6}$$

Interestingly, in (6), the posterior distributions of pre-trained SDE and optimal SDE are identical. This property is a result of the entropy-regularized term. This theorem can be generalized further.

**Lemma 4** (Bridge perseverance). *Let $\mathbb{P}^\star_{\cdot|T}(\cdot|x_T), \mathbb{P}^{\text{data}}_{\cdot|T}(\cdot|x_T)$ be distributions of $\mathbb{P}^\star, \mathbb{P}^{\text{data}}$ conditioned on states at terminal $T$, respectively. Then, $\mathbb{P}^\star_{\cdot|T}(\cdot|x_T) = \mathbb{P}^{\text{data}}_{\cdot|T}(\cdot|x_T)$.*

As an immediate corollary, we also obtain $\mathbb{P}^\star_{\cdot|0,T}(\cdot) = \mathbb{P}^{\text{data}}_{\cdot|0,T}(\cdot)$. These posterior distributions are often referred to as bridges. Note that in bridge matching methods, generative models are trained to align the bridge with the reference Brownian bridge while maintaining the initial distribution as $\nu_{\text{ini}}$ and the terminal distribution as $p_{\text{data}}$ (Shi et al., 2023). Our fine-tuning method can be viewed as a bridge-matching *fine-tuning* approach between $0$ and $T$ while keeping the terminal distribution as $\exp(r(x)/\alpha)p_{\text{data}}(x)/C_{\text{tar}}$. This bridge-matching property is valuable in preventing samples from going beyond the support of $p_{\text{data}}$.

## 6 LEARNING AN OPTIMAL INITIAL DISTRIBUTION VIA ENTROPY-REGULARIZED CONTROL

Up to this point, we have illustrated that addressing the stochastic control problem in (5) enables the creation of generative models for the target $p_{\text{tar}}$. Existing works on neural SDEs (Chen et al., 2018; Tzen and Raginsky, 2019) have established that these control problems can be effectively solved by relying on the expressive power of a neural network, and employing sufficiently small discretization steps. Although it seems plausible to employ any neural SDE solver for solving stochastic control problems (5), in typical algorithms, the initial point is fixed. Even when the initial point is unknown, it is commonly assumed to follow a Dirac delta distribution. In contrast, our control problem in Eq. (5) necessitates the learning of a stochastic initial distribution, which can function as a sampler.

A straightforward way involves assuming a Gaussian model with a mean parameterized by a neural network. While this approach is appealingly simple, it may lead to significant misspecification when $\nu^\star$ is a multi-modal distribution.

To address this challenge, we once again turn to approximating $\nu^\star$ using an SDE, as SDE-induced distributions have the capability to represent intricate multi-modal distributions. We start with a reference SDE over the interval $t \in [-T, 0]; t \in [-T, 0]; dx_t = \tilde{\sigma}(t)dw_t, x_{-T} = x_{\text{fix}}$, such that the distribution at time $0$ follows $\nu_{\text{ini}}$. Given that $\nu_{\text{ini}}$ is typically simple (e.g., $\mathcal{N}(0, \mathbb{I}_d)$), it is usually straightforward to construct such an SDE with a diffusion coefficient $\tilde{\sigma} : [0, T] \to \mathbb{R}$.

Building upon this baseline SDE, we introduce another SDE over the same interval $[-T, 0]$:

$$dx_t = q(t, x_t)dt + \tilde{\sigma}(t)dw_t, x_{-T} = x_{\text{fix}}. \tag{7}$$

---

**Algorithm 1 ELEGANT** (finE-tuning doubLe Entropy reGulArized coNTrol)

---

1: **Require**: Parameter $\alpha \in \mathbb{R}^+$, a pre-trained diffusion model with drift coefficient $f : [0, T] \times \mathbb{R}^d \to \mathbb{R}$ and diffusion coefficient $\sigma : [0, T] \to \mathbb{R}$, a base coefficient $\tilde{\sigma} : [-T, 0] \to \mathbb{R}$ and a base initial point $x_{\text{fix}}$.
2: Learn an optimal value function at $t = 0$ (i.e., $V_0^\star$) and denote it by $\hat{a} : \mathbb{R}^d \to \mathbb{R}$ invoking **Algorithm** 4 in Appeneix B.
3: Using a neural SDE solver (**Algorithm** 3), solve

$$\hat{q} = \mathrm{argmax}_q \, \mathbb{E}_{\mathbb{P}^q} \left[ \hat{a}(x_0) - \frac{\alpha}{2} \int_{-T}^0 \frac{\|q(t, x_t)\|^2}{\tilde{\sigma}^2(t)} dt \right]. \tag{9}$$

4: Let $\hat{\nu}$ be a distribution at $t = 0$ following the SDE: $dx_t = \hat{q}(t, x_t)dt + \tilde{\sigma}(t)dw_t$, $x_{-T} = x_{\text{fix}}$.
5: Using a neural SDE solver (**Algorithm** 3), solve

$$\hat{u} = \mathrm{argmax}_u \, \mathbb{E}_{\mathbb{P}^{u,\hat{\nu}}} \left[ r(x_T) - \frac{\alpha}{2} \int_{t=0}^T \frac{\|u(t, x_t)\|^2}{\sigma^2(t)} dt \right]. \tag{10}$$

6: **Output**: Drift coefficients $\hat{q}, \hat{u}$

---

**Algorithm 2 Fine-Tuned Sampler**

---

1: From $-T$ to $0$, follow the SDE: $dx_t = \hat{q}(t, x_t)dt + \tilde{\sigma}(t)dw_t$, $x_{-T} = x_{\text{fix}}$
2: From $0$ to $T$, follow the SDE: $dx_t = \{f(t, x_t) + \hat{u}(t, x_t)\}dt + \sigma(t)dw_t$.
3: **Output**: $x_T$

---

where $q : [-T, 0] \times \mathbb{R}^d \to \mathbb{R}$. This time, we aim to guide a drift coefficient $q$ over this interval $[-T, 0]$ such that the distribution at $0$ follows $\nu^\star$. Specifically, we formulate the following:

$$q^\star = \mathrm{argmax}_q \, \mathbb{E}_{\mathbb{P}^q} \left[ V_0^\star(x_0) - \frac{\alpha}{2} \int_{-T}^0 \frac{\|q(t, x_t)\|^2}{\tilde{\sigma}^2(t)} dt \right], \tag{8}$$

where $\mathbb{P}^q$ represents the measure induced by the SDE (7) with a drift coefficient $q$.

**Theorem 2** (Justification of the second control problem)**.** *The marginal density at time $0$ induced by the SDE (7) with the drift $q^\star$, i.e., $\mathbb{P}_0^{q^\star}(\cdot)$, is $\nu^\star(\cdot)$*

This shows by after learning $V_0^\star$, which will discuss in Appendix B, and solving (8) and following the learned SDE from $-T$ to $0$, we can sample from $\nu^\star$. Regarding

## 7 ALGORITHM

We are ready to present our method, **ELEGANT**, which is fully described in Algorithm 1. The algorithm consists of 3 steps:

1. Learn the value function $V_0^\star(x)$, which we will discuss in Appendix B.
2. Solve the stochastic control (9) with a neural SDE solver using the learned $V_0^\star(x)$ in 1.
3. Solve the stochastic control (10) with a neural SDE using the learned $\nu^\star$ in the second step (i.e., $\hat{\nu}$). Compared to (5), we fix the initial distribution as $\hat{\nu}$.

For our neural SDE solver, we use a standard oracle in Algorithm 3 as in Kidger et al. (2021); Chen et al. (2018) (i.e., as we use neural networks as function classes). A detailed implementation is described in Appendix E.1. To solve (9), we use the following parametrization:

$$z_t := [x_t^\top, y_t]^\top \in \mathbb{R}^{d+1}, L := -y_0 + \hat{a}(x_0), z_{\text{ini}} := x_{\text{fix}}, \bar{f} := \left[ q^\top, 0.5\alpha\|q\|^2/\tilde{\sigma}^2 \right]^\top, \bar{g} := [\tilde{\sigma}\mathbf{1}_d, 0]^\top,$$

where $y_0$ corresponds to $\int_{-T}^0 0.5\alpha\|q\|^2/\tilde{\sigma}^2 dt$. Similarly, to solve (10), we can use this solver with the following parametrization:

$$z_t := [x_t^\top, y_t]^\top \in \mathbb{R}^{d+1}, L := -y_T + r(x_T), z_{\text{ini}} := \hat{\nu}, \bar{f} := \left[ \{f + u\}^\top, 0.5\alpha\|u\|^2/\sigma^2 \right]^\top, \bar{g} := [\sigma\mathbf{1}_d, 0]^\top.$$

Finally, after learning $\hat{q}$ and $\hat{u}$, during the sampling phase, we follow the learned SDE (Algorithm 2).

---

**Algorithm 3** NeuralSDE Solver

---

1: **Input**: Diffusion coefficient $\bar{g} : [0, T] \to \mathbb{R}^{d+1}$, loss function $L : \mathbb{R}^{d+1} \to \mathbb{R}$, an initial distribution $\bar{\nu}$

2: Solve the following and denote the solution by $f^{\dagger}$:

$$f^{\dagger} = \text{argmax}_{\bar{f}:[0,T] \times \mathbb{R}^{d+1} \to \mathbb{R}^{d+1}} L(z_T), dz_t = \bar{f}(t, z_t)dt + \bar{g}(t)dw_t, z_0 \sim z_{\text{ini}}.$$

3: **Output:** $f^{\dagger}$

---

## 7.1 LIMITATION: SOURCES OF APPROXIMATION ERRORS

Lastly, we explain the factors contributing to approximation errors in our algorithm. First, our method relies on the precision of neural SDE solvers, specifically, the expressiveness of neural networks and errors from discretization (Tzen and Raginsky, 2019). Similarly, in the sampling phase, we also incur errors stemming from discretization. Additionally, our method relies on the expressiveness of another neural network in value function estimation.

As another limitation, readers might wonder about (1) computational cost of learning initial distributions, (2) memory complexity, and (3) choice of $\alpha$. We defer the discussion to Appendix 7.1.

## 8 EXPERIMENTS

We compare **ELEGANT** against several baselines across two domains. Our goal is to check that **ELEGANT** enables us to obtain diffusion models that generate high-reward samples while avoiding overoptimization and preserving diversity. We will begin by providing an overview of the baselines, describing the experimental setups, and specifying the evaluation metrics employed across all three domains. For more detailed information on each experiment, including dataset, architecture, hyperparameters, and ablation studies, refer to Appendix F.

**Methods to compare.** We compare the following:

- **ELEGANT :** Our method.
- **NO KL:** This is **ELEGANT** without the KL regularization and initial distribution learning. This essentially corresponds to AlignProp (Prabhudesai et al., 2023) and DRaFT (Clark et al., 2023) in the discrete-time formulation. While several ways to mitigate overoptimization in these papers are discussed, we will compare them with our work later in Section 8.2.
- **PPO + KL (Fan et al., 2023)** KL-penalized RL finetuning with PPO (Schulman et al., 2017) [2]
- **Guidance**: We train a reward model to predict the reward value $y$ from a sample $x$. We use this model to guide the sampling process (Dhariwal and Nichol, 2021; Graikos et al., 2022) toward high rewards. For details, refer to Appendix E.2.

**Experiment setup.** In all scenarios, we start by preparing a diffusion model with a standard dataset, containing a mix of high- and low-reward samples. Then, we create a (nominal) reward function $r$ by training a neural network reward model on a dataset with reward labels $\{x^{(i)}, r^{\star}(x^{(i)})\}_{i=1}^{n}$, ensuring that $r$ closely approximates the "genuine" reward function $r^{\star}$ on the data distribution of the dataset (i.e., on the support of pre-trained diffusion model). Following existing works (Fan et al., 2023; Black et al., 2023), we first evaluate performance in terms of $r$ in Section 8.1. However, going beyond this way, we explore an improved way to measure overoptimization in Section 8.2, 8.3.

**Evaluation.** We record the mean reward $\mathbb{E}_p[r(x_T)]$ ((b1) in Eq.(5)), the KL divergence term ((b2) in Eq.(5)). In our results, we present the mean values of the reward (Reward), the KL term (KL-Div) (and their $95\%$ confidence intervals). Our aim is to fine-tune diffusion models so that they have high (Reward) and low (KL-div): that is, to produce high-reward samples from a distribution that stays close to the data. For one of our evaluation tasks (Section 8.2), we know the true function $r^{\star}$ (though it is *not* provided to our algorithm), and therefore can directly measure the degree to which our method mitigates overoptimization by comparing the values of $r$ and $r^{\star}$ for our method and baselines.

---

[2]Note that we technically use an improved baseline elaborating on DPOK (Fan et al., 2023) and DDPO (Black et al., 2023) by directly adding a KL penalty to the rewards. For details, see Appendix E.2.

Table 1: Result for normalized GFP (Left) when we set $\alpha = 0.1$ for **ELEGANT** and **PPO + KL**. Results for TFBind (right) when we set $\alpha = 0.005$. $\pm$ means 95% confidence intervals across seeds.

(a) GFP. The pre-trained model has $0.90$ (reward), $0.0$ (KL-div)

(b) TFBind. The pre-trained model has $0.45$ (reward), $0.0$ (KL-div).

| | Reward $(r)$ ↑ | KL-Div ↓ |
|---|---|---|
| **Guidance** | $0.94 \pm 0.01$ | 624 |
| **PPO + KL** | $0.96 \pm 0.01$ | 95 |
| **ELEGANT** (Ours) | $\mathbf{0.98} \pm 0.00$ | **32** |

| | Reward $(r)$ ↑ | KL-Div ↓ |
|---|---|---|
| **Guidance** | $0.81 \pm 0.03$ | 709 |
| **PPO + KL** | $\mathbf{0.98} \pm 0.00$ | 110 |
| **ELEGANT** (Ours) | $\mathbf{0.98} \pm 0.00$ | 82 |

(a) ELEGANT (5e-3)    (b) ELEGANT (e-2)    (c) ELEGANT (5e-2)    (d) Pre-trained    (e) **NO KL**

Figure 2: Histograms of 1000 samples generated by fine-tuned diffusions for TFBind in terms of $r^{\star}(x)$ in Red and $r(x)$ in Blue. In **No KL**, the same sample with $r^{\star}$ is generated, suffering from overoptimization. **ELEGANT** can achieve both high $r$ and $r^{\star}$. The enlarged figure (a) is in Appendix F.2.3.

### 8.1 DESING OF PROTEIN AND DNA SEQUENCES

We study two distinct biological sequence tasks: GFP and TFBind (Trabucco et al., 2022). In the GFP task, $x$ represents green fluorescent protein sequences, each with a length of 237, and $r^{\star}(x)$ signifies their fluorescence value (Sarkisyan et al., 2016). In the TFBind task, $x$ represents DNA sequences, each having a length of 8, while $r^{\star}(x)$ corresponds to their binding activity with human transcription factors (Barrera et al., 2016). Using these datasets, we proceed to train transformer-based diffusion models and oracles (details in Appendix F.1).

**Results.** We present the performances in Tables 1a and 1b. It's clear that **ELEGANT** surpasses **PPO + KL** and **Guidance** in terms of rewards, maintains a smaller KL term. It's worth noting that while there is typically a tradeoff between the reward and KL term, even when fine-tuned diffusion models yield similar rewards, their KL divergences can vary significantly. This implies that, compared to **PPO + KL**, our proposal, **ELEGANT**, effectively minimizes the KL term while maintaining high rewards. This reduced KL term translates to the alleviation of overoptimization, as in Section 8.2.

Table 2: TFBind. We set $\alpha = 0.01$ for **ELEGANT** and PPO. For **Truncation**, we set $K = 0.8T$. It is seen that **ELEGANT** can circumvent overoptimization while other methods suffer from it.

| | Reward $(r)$ ↑ | Reward $(r^{\star})$ ↑ |
|---|---|---|
| **NO KL** | $\mathbf{1.0} \pm 0.0$ | $0.76 \pm 0.02$ |
| **Guidance** | $0.81 \pm 0.03$ | $0.76 \pm 0.03$ |
| **PPO + KL** | $0.987 \pm 0.001$ | $0.84 \pm 0.01$ |
| **Random** (Prabhude-sai et al., 2023) | $\mathbf{1.0} \pm 0.0$ | $0.77 \pm 0.01$ |
| **Truncation** (Clark et al., 2023) | $\mathbf{1.0} \pm 0.0$ | $0.78 \pm 0.01$ |
| **ELEGANT** (Ours) | $0.989 \pm 0.001$ | $\mathbf{0.88} \pm 0.01$ |

### 8.2 QUANTITATIVE EVALUATION OF OVEROPTIMIZATION

In TFBind, where we have knowledge of the genuine reward $r^{\star}$, we conduct a comparison between **ELEGANT** and several baselines, presented in Figure 2 and Table 2. It becomes evident that the version without KL regularization achieves high values for $r$, but not for the true reward value $r^{\star}$. In contrast, our method can overcome overoptimization by effectively minimizing the KL divergence.

In Table 2, we additionally compare algorithms that focus solely on maximizing $r(x_T)$ (i.e., ((b1) in Equation (5))) with several techniques. For instance, the approach presented in Clark et al. (2023) (DRaFT-K) can be adapted to our context by updating drift terms only in the interval $[K, T]$ rather than over the entire interval $[0, T]$ (referred to as **Truncation**). Similarly, AlignProp, as proposed by Prabhudesai et al. (2023), can be applied by randomly selecting the value of $K$ at each epoch

(referred to as **Random**). However, in the case of TFBind, it becomes evident that these techniques cannot mitigate overoptimization.

### 8.3 IMAGE GENERATION

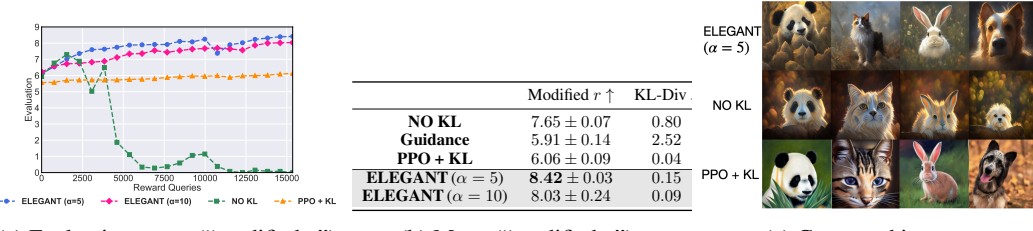

(a) Evaluation curve ("modified $r$")    (b) Mean ("modified $r$")    (c) Generated images

Figure 3: Results for fine-tuning aesthetic scores on images. Plot (a) depicts a training curve (mean of generated samples in terms of "modified $r$"). It is evident that **ELEGANT** exhibits faster training compared to **PPO + KL** while mitigating overoptimization, unlike **NO KL**. In table (b), we report the *highest* mean value of "modified $r$" across all epochs (the $x$ axis in plot (a) before $15360$ reward queries) for each method and their $95\%$ cfs. Additionally, generated images corresponding to Table (b) are provided in images (c).

Here, our goal is to fine-tune a text-to-image diffusion model to produce visually appealing pictures. We employ Stable Diffusion v1.5 as our pre-trained model (Rombach et al., 2022), a conditional diffusion model that can generate natural images given prompts (e.g., cat). In line with prior studies (Black et al., 2023; Prabhudesai et al., 2023), we use the LAION Aesthetics Predictor V2 (Schuhmann, 2022) for $r$. This predictor is a linear MLP model built on the OpenAI CLIP embeddings (Radford et al., 2021), pre-trained on a dataset over $400k$ aesthetic ratings ranging from $1$ to $10$.

**Evaluation.**    Notably, the above LAION Aesthetics Predictor V2 may not be accurate in out-of-distribution regions since it is still a learned reward function. Consequently, it may assign high scores to unnatural images that deviate far from the original prompts due to overoptimization (c.f. Figure 1). Therefore, to detect these undesirable scenarios during evaluation, we employ "modified $r$" on all generations, defined as follows: (1) querying vision language models (LLaVA in Liu et al. (2024)) to determine if images contain objects from the original prompts (e.g., cats) [3], (2) if yes, keeping the raw score $r(x)$; (3) if no, assigning a score of $0$. Notably, for all algorithms, we compute the "modified $r$" only during evaluation in order to detect overoptimization. But we don't use it during fine-tuning [4].

**Results.**    In Figure 3a and Table 3b, we present the evaluation curve and the peak number of reward queries in terms of the mean of generated samples with respect to "modified $r$". Firstly, we observe a significantly faster training speed for **ELEGANT** compared to **PPO + KL**. Secondly, comparing **ELEGANT** with **NO KL**, we notice that entropy regularization enables us to achieve higher values for "modified $r$", whereas **NO KL** begins generating images that ignore prompts early on, resulting in a rapid decline in the evaluation curve, and its peak "modified $r$" across epochs remains lower. We showcase generated images in Figure 3c, and provide additional images in Appendix F.2.3.

### 8.4 EFFECTIVENESS OF LEARNING INITIAL DISTRIBUTIONS.

Readers may want to know the effectiveness of learning the initial distribution. To address this, we also tested our algorithm without it. As shown in Table 2 (TFBinding), the result $r^\star$ is $0.86 \pm 0.01$, and in Table 3b (image generation), the modified $r$ score is $7.90 \pm 0.32$. In both cases, these values are lower than those achieved by our full algorithm, which learns the initial distribution. These findings demonstrate that learning the initial distribution is effective in mitigating overoptimization and achieving higher genuine rewards in specific practical scenarios.

## 9 CONCLUSION

We propose a theoretically and empirically grounded, computationally efficient approach for fine-tuning diffusion models. This approach helps alleviate overoptimization issues. In future work, we plan to investigate the fine-tuning of recent diffusion models more tailored for biological or chemical applications (Watson et al., 2023; Avdeyev et al., 2023; Gruver et al., 2023).

---

[3]The F1 score for detecting objects using LLaVA was $1.0$ as reported in Appendix F.2.2 and F.2.4.

[4]For results without the modificaiton on $r$, refer to Figure 6.

## REPRODUCIBILITY STATEMENT

To ensure the reproducibility of this work, we provide details for the experiments, including all the training setup, architecture in Appendix.

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

## A ADDITIONAL RELATED WORKS

In this section, we discuss additional related works.

**Fine-tuning large language models.** Much of the recent work in fine-tuning diffusion models is inspired by the wide success of fine-tuning large language models for various objectives such as instruction-following, summarization, or safety (Ouyang et al., 2022; Stiennon et al., 2020; Bai et al., 2022). Many techniques have been proposed to mitigate reward collapse in this domain, but KL regularization is the most commonly used (Gao et al., 2023). For a more comprehensive review, we direct readers to Casper et al. (2023). In our experiments, we compare to a KL-penalized RL baseline, which is analogous to the current dominant approach in language model fine-tuning.

**Sampling and control.** Control-based approaches have been extensively employed for generating samples from known unnormalized probability densities in various ways (Tzen and Raginsky, 2019; Bernton et al., 2019; Heng et al., 2020; Zhang and Chen, 2021; Berner et al., 2022; Lahlou et al., 2023; Zhang et al., 2023; Bengio et al., 2023). Notably, the most pertinent literature relates to path integral sampling (Zhang and Chen, 2021). Nevertheless, our work differs in terms of our target distribution and focus, which is primarily centered on fine-tuning. Here are more differences:

- We address how to utilize pre-trained diffusion models by properly setting rewards in the control problem.
- Their works assume an initial distribution as the Dirac delta distribution, which does not apply to diffusion models. We explore how to relax this assumption.
- We provide several proofs to show our main statement. The proof in Section C.2 is similar to that in the path integral sampler proof. However, the proofs in Sections C.1 and C.3 are novel. In particular, the proof in Section C.1 highlights the connection with bridge (flow) matching, as discussed after Lemma 4.

Our research also shares connections with path integral controls (Theodorou and Todorov, 2012; Theodorou et al., 2010; Kappen, 2007) and the concept of control as inference (Levine, 2018). However, our focus lies on the diffusion model, while their focus lies on standard RL problems.

**Markov Chain Monte Carlo (MCMC).** MCMC-based algorithms are commonly used for sampling from unnormalized densities that follow a proportionality of $\exp(r(x)/\alpha)$ (Girolami and Calderhead, 2011; Ma et al., 2019). Numerous MCMC methods have emerged, including the first-order technique referred to as MALA. The approach most closely related to incorporating MALA for fine-tuning is classifier-based guidance, as proposed in Dhariwal and Nichol (2021); Graikos et al. (2022). However, implementing classifier-based guidance is known to be unstable in practice due to the necessity of training numerous classifiers (Clark et al., 2023).

**Additional Works on Fine-Tuning Diffusion Models.** Domingo-Enrich et al. (2024) and Zhang et al. (2024) have addressed problems similar to ours, but their approaches still differ significantly. Domingo-Enrich et al. (2024) addressed the initial bias issue discussed in Section 6 by modifying the noise schedule, whereas we tackled it by introducing an additional optimization problem. Zhang et al. (2024) approached a related problem by designing an objective function inspired by a detailed balance loss in Gflownets, while our algorithm directly solves the control problem using neural SDE. Although its algorithm has certain benefits when rewards are differentiable like PPO, our primary focus is more on how to mitigate overoptimization from both theoretical and empirical perspectives. Marion et al. (2024) has explored fine-tuning in diffusion models, framing it as a bilevel optimization problem. However, they do not appear to discuss strategies for constructing objective functions, such as incorporating KL regularization to prevent overoptimization.

## B VALUE FUNCTION ESTIMATION

In the initial stage of **ELEGANT**, our objective is to learn $V_0^\star(x_0)$. To achieve this, we use

$$V_0^\star(x) = \alpha \log(\mathbb{E}_{\mathbb{P}^{\text{data}}}[\exp(r(x_T)/\alpha)|x_0 = x]),$$

which is obtained as a corollary of Lemma 1 at $t = 0$. Then, by taking a differentiable function class $\mathcal{A} : \mathbb{R}^d \to \mathbb{R}$, we use an empirical risk minimization algorithm to regress $\exp(r(x_T)/\alpha)$ on $x_0$.

While the above procedure is mathematically sound, in practice, where $\alpha$ is small, it may face numerical instability. We instead recommend the following alternative. Suppose $r(x_T) = k(x_0) + \epsilon$ where $\epsilon$ is noise under $\mathbb{P}^{\text{data}}$. Then,

$$V_0^\star(x) = k(x) + \alpha \log \mathbb{E}_{\mathbb{P}^{\text{data}}}[\exp(\epsilon/\alpha)|x_0 = x].$$

Therefore, we can directly regress $r(x_T)$ on $x_0$ since the difference between $k(x)$ and $V_0^\star(x)$ remains constant. The complete algorithm is in Algorithm 4.

---

**Algorithm 4** Optimal Value Function Estimation

---

1: **Input**: Function class $\mathcal{A} \subset [\mathbb{R}^d \to \mathbb{R}]$
2: Generate a dataset $\mathcal{D}$ that consists of pairs of $(x, y)$: $x \sim \nu_{\text{ini}}$ and $y$ as $r(x_T)$ following the pre-trained SDE: $dx_t = f(t, x_t)dt + \sigma(t)dw_t$.
3: Run an empirical risk minimization: $\hat{a} = \operatorname{argmin}_{a \in \mathcal{A}} \sum_{(x,y) \sim \mathcal{D}} \{\hat{a}(x) - y\}^2$.
4: **Output**: $\hat{a}$

---

### B.1 MORE REFINED METHODS TO LEARN VALUE FUNCTIONS

We can consider a We can directly get a pair of $x$ and $y$. Here, $\{x^{(i)}\} \sim \nu_{\text{ini}}$ and

$$\hat{y}^{(i)} := \alpha \log \hat{\mathbb{E}}_{\mathbb{P}^{\text{data}}}[\exp(r(x_T)/\alpha)|x_0 = x^{(i)}]$$

where $\hat{\cdot}$ means Monte Carlo approximation for each $x_i$. In other words,

$$\hat{\mathbb{E}}_{\mathbb{P}^{\text{data}}}[\exp(r(x_T)/\alpha)|x_0 = x_i] := \frac{1}{n} \sum_{j=1}^{n} \exp(r(x_T^{(i,j)})/\alpha)$$

where $\{r(x_T^{(i,j)})\}$ is a set of samples following $\mathbb{P}^{\text{data}}$ with initial condition: $x_0 = x_i$. Then, we are able to learn $a$ using the following ERM:

$$\hat{a} = \operatorname*{argmin}_{a \in \mathcal{A}} \sum_{i=1}^{n} \{a(x^{(i)}) - \hat{y}^{(i)}\}^2.$$

## C PROOFS

### C.1 INTUITIVE PROOF OF THEOREM 1

We first give an intuitive proof of Theorem 1.

Let $\mathbb{P}^u_{\cdot|0}(\cdot|x_0)$ be the induced distribution by the SDE:

$$dx_t = \{f(t, x_t) + u(t, x_t)\}dt + \sigma(t)dw_t.$$

over $\mathcal{C}$ conditioning on $x_0$. Similarly, let $\mathbb{P}^{\text{data}}_{\cdot|0}(\cdot|x_0)$ be the induced distribution by the SDE:

$$dx_t = f(t, x_t)dt + \sigma(t)dw_t$$

over $\mathcal{C}$ conditioning on $x_0$.

Now, we calculate the KL divergence of $\mathbb{P}^{\text{data}}_{\cdot|0}(\cdot|x_0)$ and $\mathbb{P}^u_{\cdot|0}(\cdot|x_0)$. This is equal to

$$\text{KL}(\mathbb{P}^u_{\cdot|0}(\cdot|x_0)\|\mathbb{P}^{\text{data}}_{\cdot|0}(\cdot|x_0)) = \mathbb{E}_{\{x_t\} \sim \mathbb{P}^u_{\cdot|0}(\cdot|x_0)} \left[ \int_0^T \frac{1}{2} \frac{\|u(t, x_t)\|^2}{\sigma^2(t)} dt \right]. \tag{11}$$

This is because

$$\mathrm{KL}(\mathbb{P}^u_{\cdot|0}(\cdot|x_0)\|\mathbb{P}^{\mathrm{data}}_{\cdot|0}(\cdot|x_0)) = \mathbb{E}_{\mathbb{P}^u_{\cdot|0}(\cdot|x_0)}\left[\frac{d\mathbb{P}^u_{\cdot|0}(\cdot|x_0)}{d\mathbb{P}^{\mathrm{data}}_{\cdot|0}(\cdot|x_0)}\right]$$

$$= \mathbb{E}_{\mathbb{P}^u_{\cdot|0}(\cdot|x_0)}\left[\int_0^T \frac{1}{2}\frac{\|u(t,x_t)\|^2}{\sigma^2(t)}dt + \int_0^T u(t,x_t)dw_t\right]$$

(Girsanov theorem)

$$= \mathbb{E}_{\mathbb{P}^u_{\cdot|0}(\cdot|x_0)}\left[\int_0^T \frac{1}{2}\frac{\|u(t,x_t)\|^2}{\sigma^2(t)}dt\right].$$

(Martingale property of Itô integral)

Therefore, the objective function in (5) is equivalent to

$$\mathrm{obj} = \mathbb{E}_{\mathbb{P}^{u,\nu}}[r(x_T)] - \alpha\mathrm{KL}(\mathbb{P}^{u,\nu}\|\mathbb{P}^{\mathrm{data}}). \tag{12}$$

This is because

$$\mathbb{E}_{\mathbb{P}^{u,\nu}}[r(x_T)] - \alpha\mathrm{KL}(\nu\|\nu_{\mathrm{ini}}) - \alpha\mathbb{E}_{\mathbb{P}^{u,\nu}}\left[\int_0^T \frac{1}{2}\frac{\|u(t,x_t)\|^2}{\sigma^2(t)}dt\right]$$

$$= \mathbb{E}_{\mathbb{P}^{u,\nu}}[r(x_T)] - \alpha\mathrm{KL}(\nu\|\nu_{\mathrm{ini}}) - \alpha\mathbb{E}_{x_0\sim\nu}\left[\mathrm{KL}(\mathbb{P}^u_{\cdot|0}(\cdot|x_0)\|\mathbb{P}^{\mathrm{data}}_{\cdot|0}(\cdot|x_0))\right]$$

$$= \mathbb{E}_{\mathbb{P}^{u,\nu}}[r(x_T)] - \alpha\mathrm{KL}(\mathbb{P}^{u,\nu}\|\mathbb{P}^{\mathrm{data}}).$$

The objective function is further changed as follows:

$$\mathrm{obj} = \mathbb{E}_{\mathbb{P}^{u,\nu}}[r(x_T)] - \alpha\mathrm{KL}(\mathbb{P}^{u,\nu}\|\mathbb{P}^{\mathrm{data}})$$

$$= \underbrace{\mathbb{E}_{x_T\sim\mathbb{P}^{u,\nu}_T}[r(x_T)] - \alpha\mathrm{KL}(\mathbb{P}^{u,\nu}_T\|\mathbb{P}^{\mathrm{data}}_T)}_{\text{Term (a)}} - \underbrace{\alpha\mathbb{E}_{x_T\sim\mathbb{P}^{u,\nu}_T}[\mathrm{KL}(\mathbb{P}^{u,\nu}_T(\tau|x_T)\|\mathbb{P}^{\mathrm{data}}_T(\tau|x_T))]}_{\text{Term (b)}}\}.$$

By optimizing (a) and (b) over $\mathbb{P}^{u,\nu}$, we get

$$\mathbb{P}^\star_T(x_T) = \exp(r(x_T)/\alpha)\mathbb{P}^{\mathrm{data}}(x_T)/C,$$

$$\mathbb{P}^\star_T(\tau|x_T) = \mathbb{P}^{\mathrm{data}}_T(\tau|x_T). \tag{13}$$

Hence, we have

$$\mathbb{P}^\star(\tau) = \mathbb{P}^\star_T(x_T) \times \mathbb{P}^\star_T(\tau|x_T) = \frac{\exp(r(x_T)/\alpha)\mathbb{P}^{\mathrm{data}}(\tau)}{C}.$$

**Remark 1.** *Some readers might wonder in the part we optimize over $\mathbb{P}_{f,\nu}$ rather than $f,\nu$. Indeed, this step would go through when we use non-Markovian drifts for $f$. While we use Markovian drift, this part still goes through because the optimal drift needs to be known as Markovian anyway. We choose to present this proof first because it can more clearly convey our message of bridge preserving property in (13). We will formalize it in Section C.2 and Section C.3.*

## C.2 FORMAL PROOF OF THEOREM 1

Firstly, we aim to show that the optimal conditional distribution over $\mathcal{C}$ on $x_0$ (i.e., $\mathbb{P}^{u^\star}_{\cdot|0}(\tau|x_0)$) is equivalent to

$$\frac{\mathbb{P}^{\mathrm{data}}_{\cdot|0}(\tau|x_0)\exp(r(x_T)/\alpha)}{C(x_0)}, \quad C(x_0) := \exp(V^\star_0(x)/\alpha).$$

To do that, we need to check that the above is a valid distribution first. This is indeed valid because the above is decomposed into

$$\underbrace{\frac{\exp(r(x_T)/\alpha)\mathbb{P}^{\mathrm{data}}(x_T|x_0)}{C(x_0)}}_{(\alpha 1)} \times \underbrace{\mathbb{P}^{\mathrm{data}}_{\cdot|0}(\tau|x_0,x_T)}_{(\alpha 2)}, \tag{14}$$

and both $(\alpha 1)$, $(\alpha 2)$ are valid distributions. Especially, for the term $(\alpha 1)$, we can check as follows:

$$C(x_0) = \int \exp(r(x_T)/\alpha) d\mathbb{P}_{T|0}^{\text{data}}(x_T|x_0)) = \mathbb{E}_{\mathbb{P}_{\cdot|x_0}^{\text{data}}}[\exp(r(x_T)/\alpha)] = \exp(V_0^\star(x)/\alpha).$$

(Use Lemma 1)

Now, after checking (14) is a valid distribution, we calculate the KL divergence:

$$\text{KL}\left(\mathbb{P}_{\cdot|0}^{u^\star}(\tau|x_0)\middle\|\frac{\mathbb{P}_{\cdot|0}^{\text{data}}(\tau|x_0)\exp(r(x_T)/\alpha)}{C(x_0)}\right)$$

$$= \text{KL}(\mathbb{P}_{\cdot|0}^{u^\star}(\tau|x_0)\|\mathbb{P}_{\cdot|0}^{\text{data}}(\tau|x_0)) - \mathbb{E}_{\mathbb{P}_{\cdot|0}^{u^\star}(\cdot|x_0)}\left[r(x_T)/\alpha - \log C(x_0)|x_0\right]$$

$$= \mathbb{E}_{\mathbb{P}_{\cdot|0}^{u^\star}(\cdot|x_0)}\left[\left\{\int_0^T \frac{1}{2}\frac{\|u^\star(t,x_t)\|^2}{\sigma^2(t)}\right\}dt - r(x_T)/\alpha + \log C(x_0) \mid x_0\right] \qquad \text{(Use (11))}$$

$$= -V_0^\star(x_0)/\alpha + \log C(x_0). \qquad\qquad \text{(Definition of optimal value function)}$$

Therefore,

$$\text{KL}\left(\mathbb{P}_{\cdot|0}^{u^\star}(\tau|x_0)\|\frac{\mathbb{P}_{\cdot|0}^{\text{data}}(\tau|x_0)\exp(r(x_T)/\alpha)}{C(x_0)}\right) = -V_0^\star(x_0)/\alpha + \log C(x_0) = 0.$$

Hence,

$$\mathbb{P}_{\cdot|0}^{u^\star}(\tau|x_0) = \frac{\mathbb{P}_{\cdot|0}^{\text{data}}(\tau|x_0)\exp(r(x_T)/\alpha)}{C(x_0)}.$$

Now, we aim to calculate an exact formulation of the optimal initial distribution. We just need to solve

$$\underset{\nu'}{\arg\max}\int V_0^\star(x)\nu'(x) - \alpha\text{KL}(\nu'\|\nu_{\text{ini}}).$$

The closed-form solution is

$$\exp(V_0^\star(x)/\alpha)\nu_{\text{ini}}(x)/C$$

where $C := \int \exp(V_0^\star(x)/\alpha)\nu_{\text{ini}}(x)dx$.

Combining all together, we have been proved that the induced trajectory by the optimal control and the optimal initial distribution is

$$\mathbb{P}_{\cdot|0}^{u^\star}(\tau|x_0) = \frac{\mathbb{P}_{\cdot|0}^{\text{data}}(\tau|x_0)\exp(r(x_T)/\alpha)}{C(x_0)}, \quad \nu^\star \sim \frac{C(x_0)\nu_{\text{ini}}(x_0)}{C}.$$

Therefore,

$$\mathbb{P}^{u^\star,\nu^\star}(\tau) = \mathbb{P}_{\cdot|0}^{u^\star}(\tau|x_0)\nu^\star(x_0) = \frac{\mathbb{P}_{\cdot|0}^{\text{data}}(\tau|x_0)\exp(r(x_T)/\alpha)}{C(x_0)} \times \frac{C(x_0)\nu_{\text{ini}}(x_0)}{C}$$

$$= \frac{\mathbb{P}_{\cdot|0}^{\text{data}}(\tau|x_0)\exp(r(x_T)/\alpha)\nu_{\text{ini}}(x_0)}{C} = \frac{\mathbb{P}^{\text{data}}(\tau)\exp(r(x_T)/\alpha)}{C}.$$

**Marginal distribution at $t$.** Finally, consider the marginal distribution at $t$. By marginalizing before $t$, we get

$$\mathbb{P}^{\text{data}}(\tau_{[t,T]}) \times \exp(r(x_T)/\alpha)/C.$$

Next, by marginalizing after $t$,

$$\mathbb{P}_t^{\text{data}}(x)/C \times \mathbb{E}_{\mathbb{P}_{\text{data}}}[\exp(r(x_T)/\alpha)|x_t = x].$$

Using Feynman–Kac formulation in Lemma 1, this is equivalent to

$$\mathbb{P}_t^{\text{data}}(x)\exp(V_t^\star(x)/\alpha)/C.$$

**Marginal distribution at $T$.**   We marginalize before $T$. We have the following

$$\mathbb{P}_T^{\text{data}}(x)\exp(r(x)/\alpha)/C.$$

## C.3   ANOTHER FORMAL PROOF OF THEOREM 1

First, noting the loss in (5) becomes

$$\mathbb{E}_{x_0\sim\nu}[V_0^\star(x_0) - \alpha\text{KL}(\nu(x_0)/\nu_{\text{ini}}(x_0))],$$

by optimizing over $\nu\in\Delta(\mathcal{X})$, we can easily prove that the optimal initial distribution is

$$\exp\left(\frac{V_0^\star(x)}{\alpha}\right)\nu_{\text{ini}}(x)/C.$$

Hereafter, our goal is to prove that the marginal distribution at $t$ (i.e., $\mathbb{P}_t^\star$) is indeed $g_t(x)$ defined by

$$g_t(x) := \exp\left(\frac{V_t^\star(x)}{\alpha}\right)\mathbb{P}_t^{\text{data}}(x)/C$$

Using Lemma 1, we can show that the SDE with the optimal drift term is

$$dx_t = \left\{f(t,x) + \frac{\sigma^2(t)}{\alpha}\nabla V_t^\star(x)\right\}dt + \sigma(t)dw_t.$$

Then, what we need to prove is that the density $g_t\in\Delta(\mathbb{R}^d)$ satisfies the Kolmogorov forward equation:

$$\frac{g_t(x)}{dt} + \sum_i \frac{d}{dx^{[i]}}\left[\left\{f^{[i]}(t,x) + \frac{\sigma^2(t)}{\alpha}\nabla_{x^{[i]}}V_t^\star(x)\right\}g_t(x)\right] - \frac{\sigma^2(t)}{2}\sum_i\frac{d^2g_t(x)}{dx^{[i]}dx^{[i]}} = 0 \quad (15)$$

where $f = [f^{[1]},\cdots,f^{[d]}]^\top$. Indeed, this (15) is proved as follows:

$$\frac{dg_t(x)}{dt} + \sum_i\frac{d}{dx^{[i]}}\left[\left\{f^{[i]}(t,x) + \frac{\sigma^2(t)}{\alpha}\nabla_{x^{[i]}}V_t^\star(x)\right\}g_t(x)\right] - \frac{\sigma^2(t)}{2}\sum_i\frac{d^2g_t(x)}{dx^{[i]}dx^{[i]}}$$

$$= \frac{1}{C}\exp\left(\frac{V_t^\star(x)}{\alpha}\right)\left\{\frac{d\mathbb{P}_t^{\text{data}}(x)}{dt} + \sum_i\nabla_{x^{[i]}}(\mathbb{P}_t^{\text{data}}(x)f^{[i]}(t,x)) - \frac{\sigma^2(t)}{2}\sum_i\frac{d^2\mathbb{P}_t^{\text{data}}(x)}{dx^{[i]}dx^{[i]}}\right\}$$

$$+ \frac{1}{C}\mathbb{P}_t^{\text{data}}(x)\left\{\frac{d\exp(V_t^\star(x)/\alpha)}{dt} + \sum_i f^{[i]}(t,x)\nabla_{x^{[i]}}(\exp(V_t^\star(x)/\alpha)) - \frac{\sigma^2(t)}{2}\sum_i\frac{d^2\exp(V_t^\star(x)/\alpha)}{dx^{[i]}dx^{[i]}}\right\}$$

$$+ \frac{1}{C}\mathbb{P}_t^{\text{data}}(x)\times\frac{\sigma^2(t)}{2}\sum_i\frac{d^2\exp(V_t^\star(x)/\alpha)}{dx^{[i]}dx^{[i]}}$$

$$= 0 + 0.$$

Note in the final step, we use

$$\frac{d\mathbb{P}_t^{\text{data}}(x)}{dt} + \sum_i\nabla_{x^{[i]}}(\mathbb{P}_t^{\text{data}}(x)f^{[i]}(t,x)) - \frac{\sigma^2(t)}{2}\sum_i\frac{d^2\mathbb{P}_t^{\text{data}}(x)}{dx^{[i]}dx^{[i]}} = 0,$$

which is derived from the Kolmogorov forward equation, and the optimal value function satisfies the following

$$\frac{\sigma^2(t)}{2}\sum_i\frac{d^2\exp(V_t^\star(x)/\alpha)}{dx^{[i]}dx^{[i]}} + f\cdot\nabla\exp(V_t^\star(x)/\alpha) + \frac{d\exp(V_t^\star(x)/\alpha)}{dt} = 0,$$

which will be shown in the proof of Lemma 1 as in (18).

Hence, (15) is proved, and $g_t$ is $d\mathbb{P}_t^\star/d\mu$.

### C.4  PROOF OF LEMMA 1

From the Hamilton–Jacobi–Bellman (HJB) equation, we have

$$\max_u \left\{ \frac{\sigma^2(t)}{2} \sum_i \frac{d^2 V_t^\star(x)}{dx^{[i]} dx^{[i]}} + \{f + u\} \cdot \nabla V_t^\star(x) + \frac{dV_t^\star(x)}{dt} - \frac{\alpha \|u\|_2^2}{2\sigma^2(t)} \right\} = 0. \qquad (16)$$

where $x^{[i]}$ is a $i$-th element in $x$. Hence, by simple algebra, we can prove that the optimal control satisfies

$$u^\star(t, x) = \frac{\sigma^2(t)}{\alpha} \nabla V_t^\star(x).$$

By plugging the above into the HJB equation (16), we get

$$\frac{\sigma^2(t)}{2} \sum_i \frac{d^2 V_t^\star(x)}{dx^{[i]} dx^{[i]}} + f \cdot \nabla V_t^\star(x) + \frac{dV_t^\star(x)}{dt} + \frac{\sigma^2(t)\|\nabla V_t^\star(x)\|_2^2}{2\alpha} = 0, \qquad (17)$$

which characterizes the optimal value function. Now, using (17), we can show

$$\frac{\sigma^2(t)}{2} \sum_i \frac{d^2 \exp(V_t^\star(x)/\alpha)}{dx^{[i]} dx^{[i]}} + f \cdot \nabla \exp(V_t^\star(x)/\alpha) + \frac{d \exp(V_t^\star(x)/\alpha)}{dt}$$

$$= \exp\left(\frac{V_t^\star(x)}{\alpha}\right) \times \left\{ \frac{\sigma^2(t)}{2} \sum_i \frac{d^2 V_t^\star(x)}{dx^{[i]} dx^{[i]}} + f \cdot \nabla V_t^\star(x) + \frac{dV_t^\star(x)}{dt} + \frac{\sigma^2(t)\|\nabla V_t^\star(x)\|_2^2}{2\alpha} \right\}$$

$$= 0.$$

Therefore, to summarize, we have

$$\frac{\sigma^2(t)}{2} \sum_i \frac{d^2 \exp(V_t^\star(x)/\alpha)}{dx^{[i]} dx^{[i]}} + f \cdot \nabla \exp(V_t^\star(x)/\alpha) + \frac{d \exp(V_t^\star(x)/\alpha)}{dt} = 0, \qquad (18)$$

$$V_T^\star(x) = r(x). \qquad (19)$$

Finally, by invoking the Feynman-Kac formula (Shreve et al., 2004), we obtain the conclusion:

$$\exp\left(\frac{V_t^\star(x)}{\alpha}\right) = \mathbb{E}_{\mathbb{P}\text{data}} \left[ \exp\left(\frac{r(x_T)}{\alpha}\right) | x_t = x \right].$$

### C.5  PROOF OF LEMMA 2

Recall $u^\star(t, x) = \frac{\sigma^2(t)}{\alpha} \times \nabla_x V_t^\star(x)$ from the proof of Lemma 1. Then, we have

$$u^\star(t, x) = \frac{\sigma^2(t)}{\alpha} \times \nabla_x V_t^\star(x) = \frac{\sigma^2(t)}{\alpha} \times \alpha \frac{\nabla_x \exp(V_t^\star(x)/\alpha)}{\exp(V_t^\star(x)/\alpha)}$$

$$= \sigma^2(t) \times \frac{\nabla_x \mathbb{E}_{\mathbb{P}\text{data}}[\exp(r(x_T)/\alpha)|x_t = x]}{\mathbb{E}_{\mathbb{P}\text{data}}[\exp(r(x_T)/\alpha)|x_t = x]}.$$

### C.6  PROOF OF LEMMA 3

Recall

$$\mathbb{P}^\star(\tau) = \mathbb{P}^\text{data}(\tau) \exp(r(x_T)/\alpha)/C$$

By marginalizing before $s$, we get

$$\mathbb{P}^\star_{[s,T]}(\tau_{[s,T]}) \times \exp(r(x_T)/\alpha)/C$$

Next, marginalizing after $t$,

$$\mathbb{P}^\text{data}_{[s,t]}(\tau_{[s,t]}) \times \mathbb{E}_{\mathbb{P}\text{data}}[\exp(r(x_T)/\alpha)|x_t]/C$$

$$= \mathbb{P}^\text{data}_{[s,t]}(\tau_{[s,t]}) \exp(V_t^\star(x_t)/\alpha)/C.$$

Finally, by marginalizing between $s$ and $t$, the joint distribution at $(s, t)$ is

$$\mathbb{P}^\text{data}_{s,t}(x_s, x_t) \exp(V_t^\star(x_t)/\alpha)/C.$$

**Forward conditional distribution.**

$$\mathbb{P}^{\star}_{t|s}(x_t|x_s) = \frac{\mathbb{P}^{\text{data}}_{s,t}(x_s, x_t) \exp(V^{\star}_t(x_t)/\alpha)/C}{\mathbb{P}^{\text{data}}_s(x_s) \exp(V^{\star}_s(x_s)/\alpha)/C} = \mathbb{P}^{\text{data}}_{s|t}(x_t|x_s) \frac{\exp(V^{\star}_t(x_t)/\alpha)}{\exp(V^{\star}_s(x_s)/\alpha)}.$$

**Backward conditional distribution.**

$$\mathbb{P}^{\star}_{s|t}(x_s|x_t) = \frac{\mathbb{P}^{\text{data}}_{s,t}(x_s, x_t) \exp(V^{\star}_t(x_t)/\alpha)/C}{\mathbb{P}^{\text{data}}_s(x_t) \exp(V^{\star}_t(x_t)/\alpha)/C} = \mathbb{P}^{\text{data}}_{s|t}(x_s|x_t).$$

### C.7 PROOF OF LEMMA 4

Use a statement in (13) the proof of Theorem 1.

### C.8 PROOF OF THEOREM 2

We omit the proof since it is the same as the proof of Theorem 1.

## D DIFFUSION MODELS

In this section, we provide an overview of continuous-time generative models. The objective is to train a SDE in such a way that the marginal distribution at time $T$ follows $p_{\text{data}}$. While $p_{\text{data}}$ is not known, we do have access to a dataset that follows this distribution.

**Denoising diffusion models** DDPMs (Song et al., 2020) are a widely adopted class of generative models. We start by considering a forward stochastic differential equation (SDE) represented as:

$$d\mathbf{y}_t = -0.5\mathbf{y}_t dt + dw_t, \mathbf{y}_0 \sim p_{\text{data}}, \tag{20}$$

defined on the time interval $[0, T]$. As $T$ tends toward infinity, the limit of this distribution converges to $\mathcal{N}(0, \mathbb{I}_d)$, where $\mathbb{I}_d$ denotes a $d$-dimensional identity matrix. Let $\mathbb{Q}$ be a measure on $\mathcal{C}$ induced by the forward SDE (20). Consequently, a generative model can be defined using its time-reversal: $x_t = \mathbf{y}_{T-t}$, which is characterized by:

$$dx_t = \{0.5x_t - \nabla \log \mathbb{Q}_{T-t}(x_t)\}dt + dw_t, \ x_0 \sim \mathcal{N}(0, \mathbb{I}_d).$$

A core aspect of DDPMs involves learning the score $\nabla \log \mathbb{Q}_t$ by optimizing the following loss with respect to $S$:

$$\mathbb{E}_{\mathbb{Q}}[\|\nabla \log \mathbb{Q}_{t|0}(x_t|x_0) - S(t, x_t)\|_2^2].$$

A potential limitation of the above approach is that the forward SDE (20) might not converge to a predefined prior distribution, such as $\mathcal{N}(0, \mathbb{I}_d)$, within a finite time $T$. To address this concern, we can employ the Schrödinger Bridge formulation (De Bortoli et al., 2021).

**Diffusion Schrödinger Bridge.** A potential bottleneck of the diffusion model is that the forward SDE might not converge to a pre-specified prior $\mathcal{N}(0, \mathbb{I}_d)$ with finite $T$. To mitigate this problem, De Bortoli et al. (2021) proposed the following Diffusion Schrödinger Bridge. Being inspired by Schrödinger Bridge formulation (Schrödinger, 1931) they formulate the problem:

$$\underset{\mathbb{P}}{\arg\min} \, \text{KL}(\mathbb{P}\|\mathbb{P}_{\text{ref}}) \, \text{s.t.} \mathbb{P}_0 = \nu_{\text{ini}}, \mathbb{P}_T = p_{\text{data}}.$$

where $\mathbb{P}_{\text{ref}}$ is a reference distribution on $\mathcal{C}$ such as a Wiener process, $\mathbb{P}_0$ is a margin distribution of $\mathbb{P}$ at time 0, and $\mathbb{P}_T$ is similarly defined. To solve this problem, De Bortoli et al. (2021) proposed an iterative proportional fitting, which is a continuous extension of the Sinkhorn algorithm (Cuturi, 2013), while learning the score functions from the data as in DDM.

**Bridge Matching.** In DDPMs, we have formulated a generative model based on the time-reversal process and aimed to learn a score function $\nabla \log \mathbb{Q}_t$ from the data. Another recent popular approach involves emulating the reference Brownian bridge given $0, T$ with fixed a pre-defined initial distribution $\nu_{\text{ini}}$ and a data distribution $p_{\text{data}}$ at time $T$ (Shi et al., 2023; Liu et al., 2022). To elaborate further, let's begin by introducing a reference SDE:

$$d\bar{x}_t = \sigma(t)dw_t, \quad \bar{x}_0 \sim \nu_{\text{ini}}.$$

Here, we overload the notation with $\mathbb{Q}$ to indicate an induced SDE. The Brownian bridge $\mathbb{Q}_{\cdot|[0,T]}(\cdot|x_0, x_T)$ is defined as:

$$d\bar{x}_t^{0,T} = \sigma(t)^2 \nabla \log \mathbb{Q}_{T|t}(x_T|\bar{x}_t^{0,T}) + \sigma(t)dw_t, x_0^{0,T} = x_0,$$

where $x_T^{0,T} = x_T$. The explicit calculation of this Brownian bridge is given by:

$$\nabla \log \mathbb{Q}_{T|t}(x_T|\bar{x}_t^{0,T}) = \frac{x_T - x_t}{T - t}.$$

Now, we define a target SDE as follows:

$$dx_t = f(t, x_t)dt + \sigma(t)dw_t, \quad x_0 \sim \nu_{\text{ini}}.$$

This SDE aims to induce the same Brownian bridge as described earlier and should generate the distribution $p_{\text{data}}$ at time $T$. We use $\mathbb{P}$ to denote the measure induced by this SDE. While the specific drift term $f$ is unknown, we can sample any time point $t(0 < t < T)$ from $\mathbb{P}$ by first sampling $x_0$ and $x_T$ from $\nu_{\text{ini}}$ and $p_{\text{data}}$, respectively, and then sampling from the reference bridge, i.e., the Brownian bridge. To learn this drift term $f$, we can use the following characterization:

$$f(t, x_t) = \mathbb{E}_{\mathbb{P}}[\nabla \log \mathbb{Q}_{T|t}(x_T|x_t)|x_t = x_t].$$

Subsequently, the desired drift term can be learned using the following loss function with respect to $f$:

$$\mathbb{E}_{\mathbb{P}}[\|\nabla \log \mathbb{Q}_{T|t}(x_T|x_t) - f(t, x_t)\|^2].$$

Bridge matching can be formulated in various equivalent ways, and for additional details, we refer the readers to Shi et al. (2023); Liu et al. (2022).

# E   DETAILS OF IMPLEMENTATION

## E.1   IMPLEMENTATION DETAILS OF NEURAL SDE

We aim to solve

$$\underset{\bar{f}:[0,T]\times\mathcal{Z}\to\mathcal{Z}}{\text{argmax}} \; L(z_T), dz_t = \bar{f}(t, z_t)dt + \bar{g}(t)dw_t, z_0 \sim \bar{\nu}.$$

Here is a simple method we use. Regarding more details, refer to Kidger et al. (2021); Chen et al. (2018).

Suppose that $\bar{f}(\cdot; \theta)$ is parametrized by $\theta$. Then, we update this $\theta$ with SGD. Consider at iteration $j$. Fix $\theta_j$ in $\bar{f}(t, z_t; \theta_j)$. Then, by simulating an SDE with

$$dz_t = \bar{f}(t, z_t; \theta)dt + \bar{g}(t)dw_t, z_0 \sim \bar{\nu},$$

we obtain $N$ trajectories

$$\{z_0^{(i)}, \cdots, z_T^{(i)}\}_{i=1}^N.$$

In this step, we are able to use any off-the-shelf discretization methods. For example, starting from $z_0^{(i)} \sim \nu$, we are able to obtain a trajectory as follows:

$$z_t^{(i)} = z_{t-1}^{(i)} + \bar{f}(t-1, z_{t-1}^{(i)}; \theta)\Delta t + \bar{g}(t-1)\Delta w_t, \quad \Delta w_t \sim \mathcal{N}(0, (\Delta t)^2).$$

Finally, using automatic differentiation, we update $\theta$ with the following:

$$\theta_{j+1} = \theta_j - \rho \nabla_\theta \left\{ \frac{1}{N} \sum_{i=1}^N L(z_T^{(i)}) \right\} \Big|_{\theta=\theta_j}$$

where $\rho$ is a learning rate. We use Adam in this step for the practical selection of the learning rate $\rho$.

### E.2 BASELINES

We consider the following baselines.

**PPO + KL.** Considering the discretized formulation of diffusion models (Black et al., 2023; Fan et al., 2023), we use the following update rule:

$$\nabla_\theta \mathbb{E}_\mathcal{D} \sum_{t=1}^T \left[ \min \left\{ \tilde{r}_t(x_0, x_t) \frac{p(x_t|x_{t-1};\theta)}{p(x_t|x_{t-1};\theta_{\text{old}})}, \tilde{r}_t(x_0, x_t) \cdot \text{Clip} \left( \frac{p(x_t|x_{t-1};\theta)}{p(x_t|x_{t-1};\theta_{\text{old}})}, 1 - \epsilon, 1 + \epsilon \right) \right\} \right],$$

(21)

$$\tilde{r}_t(x_0, x_t) = -r(x_T) + \alpha \underbrace{\frac{\|u(t, x_t; \theta)\|^2}{2\sigma^2(t)}}_{\text{KL term}}, \quad p(x_t|x_{t-1};\theta) = \mathcal{N}(u(t, x_t; \theta) + f(t, x_t), \sigma(t)) \quad (22)$$

where $f(t, x_t)$ is a pre-trained drift term and $\theta$ is a parameter to optimize.

Note that DPOK (Fan et al., 2023) uses the following update:

$$\nabla_\theta \mathbb{E}_\mathcal{D} \sum_{t=1}^T \left[ \min \left\{ -r(x_0) \frac{p(x_t|x_{t-1};\theta)}{p(x_t|x_{t-1};\theta_{\text{old}})}, -r(x_0) \cdot \text{Clip} \left( \frac{p(x_t|x_{t-1};\theta)}{p(x_t|x_{t-1};\theta_{\text{old}})}, 1 - \epsilon, 1 + \epsilon \right) \right\} + \alpha \underbrace{\frac{\|u(t, x_t; \theta)\|^2}{2\sigma^2(t)}}_{\text{KL term}} \right]$$

where the KL term is directly differentiated. We did not use the DPOK update rule because DDPO appears to outperform DPOK even without a KL penalty (Black et al. (2023), Appendix C), so we implemented this baseline by modifying the DDPO codebase to include the added KL penalty term (Equation (22)).

**Guidance.** We use the following implementation of guidance (Dhariwal and Nichol, 2021):

- For each $t \in [0, T]$, we train a model: $\mathbb{P}_t(y|x_t)$ where $x_t$ is a random variable induced by the pre-trained diffusion model.
- We fix a guidance level $\gamma \in \mathbb{R}_{>0}$, target value $y_{\text{con}} \in \mathbb{R}$, and at inference time (during each sampling step), we use the following score function

$$\nabla_x \log \mathbb{P}_t(x|y = y_{\text{con}}) = \nabla_x \log \mathbb{P}_t(x) + \gamma \nabla_x \log \mathbb{P}_t(y = y_{\text{con}}|x).$$

A remaining question is how to model $p(y|x)$. In our case, for the biological example, we make a label depending on whether $x$ is top 10% or not and train a binary classifier. In image experiments, we construct a Gaussian model: $p(y|x) = \mathcal{N}(y - \mu_\theta(x), \sigma^2)$ where $y$ is the reward label, $\mu_\theta$ is the reward model we need to train, and $\sigma$ is a fixed hyperparameter.

## F EXPERIMENT DETAILS

### F.1 DETAILS FOR TASKS IN BIOLOGICAL SEQUENCES

#### F.1.1 DATASET.

**TFBind8.** The number of original dataset size is 65792. Each data consists of a DNA sequence with 8-length. We represent each data as a one-hot encoding vector with dimension $8 \times 4$. To construct diffusion models, we use all datasets. We use half of the dataset to construct a learned reward $r$ to make a scenario where oracles are imperfect.

**GFP.** The original dataset contains $56,086$ data points, each comprising an amino acid sequence with a length of 237. We represent each data point using one-hot encoding with a dimension of $237 \times 20$. Specifically, we model the difference between the original sequence and the baseline sequence. For our experiments, we selected the top $33,637$ samples following (Trabucco et al., 2022) and trained diffusion models and oracles using this selected dataset.

Table 3: Architecture of diffusion models for TFBind

| Layer | Input Dimension | Output dimension | Explanation |
|---|---|---|---|
| 1 | $1\ (t)$ | $256\ (t')$ | Get time feature |
| 1 | $8 \times 4\ (x)$ | $64\ (x')$ | Get positional encoder (Denote $x'$) |
| 2 | $8 \times 4 + 256 + 64\ (x, t, x')$ | $64\ (\bar{x})$ | Transformer encoder |
| 3 | $64\ (\bar{x})$ | $8 \times 4\ (x)$ | Linear |

Table 4: Architecture of oracles for TFBind

| | Input dimension | Output dimension | Explanation |
|---|---|---|---|
| 1 | $8 \times 4$ | 500 | Linear |
| 1 | 500 | 500 | ReLU |
| 2 | 500 | 200 | Linear |
| 2 | 200 | 200 | ReLU |
| 3 | 200 | 1 | Linear |
| 3 | 200 | 1 | ReLU |
| 4 | 1 | 1 | Sigmoid |

Table 5: Primary hyperparameters for fine-tuning. For all methods, we use the Adam optimizer.

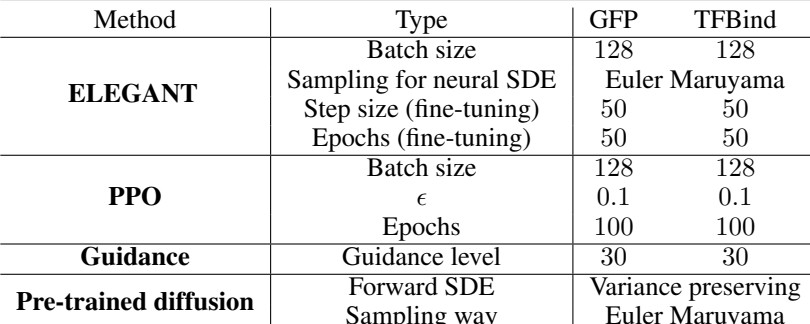

| Method | Type | GFP | TFBind |
|---|---|---|---|
| **ELEGANT** | Batch size | 128 | 128 |
| | Sampling for neural SDE | Euler Maruyama | |
| | Step size (fine-tuning) | 50 | 50 |
| | Epochs (fine-tuning) | 50 | 50 |
| **PPO** | Batch size | 128 | 128 |
| | $\epsilon$ | 0.1 | 0.1 |
| | Epochs | 100 | 100 |
| **Guidance** | Guidance level | 30 | 30 |
| **Pre-trained diffusion** | Forward SDE | Variance preserving | |
| | Sampling way | Euler Maruyama | |

### F.1.2 STRUCTURE OF NEURAL NETWORKS.

We describe the implementation of neural networks in more detail.

**Diffusion models and fine-tuning.** For diffusion models in TFBind, we use a neural network to model score functions in Table 3. We use a similar network for the GFP dataset and fine-tuning parts.

**Oracles to obtain score functions.** To construct oracles in TFBind, we employ the neural networks listed in Table 4. For GFP, we utilize a similar network.

### F.1.3 HYPERPARAMETERS

We report a set of important hyperparameters in Table 5.

### F.1.4 ADDITIONAL RESULTS

In this section, we add additional results to support the main paper.

**Enlarged figure of ELEGANT (0.005).** We add the enlarged figure of **ELEGANT** (0.005) in Table 2.

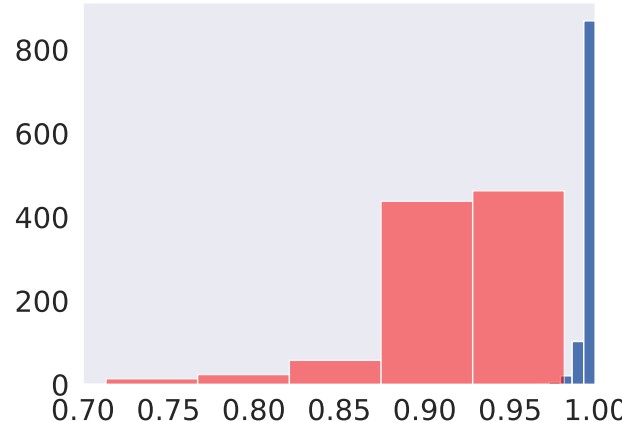

Figure 4: Enlarged histogram of **ELEGANT** (0.005) in Table 2.

Table 6: TFBind. We set $\alpha = 0.01$ for **ELEGANT** and PPO.

|  | Reward $(r) \uparrow$ | Reward $(r^\star) \uparrow$ |
|---|---|---|
| **DDPO** | $0.99 \pm 0.0$ | $0.80 \pm 0.02$ |
| **DPOK** | $0.99 \pm 0.00$ | $0.78 \pm 0.02$ |
| **PPO + KL** | $0.99 \pm 0.0$ | $0.84 \pm 0.01$ |

**Comparison of PPO+KL, DPOK, DDPO.** We compare **PPO+KL**, **DPOK**, and **DPPO** in Table 6 on page 25.

Note with respect to DPOK and DDPO, we use the term "PPO + KL" to convey that our "PPO+KL" updates both the reward term and KL term with PPO, whereas the original DPOK optimizes the reward term using PPO but employs a non-PPO approach for optimizing the KL term. We have observed that "PPO + KL" yields more stable optimization compared to the precise algorithm in DPOK, and aligns with a more conventional optimization method in the RL community.

**Ablation studies in terms of $\alpha$.** We have also conducted ablation studies with varying hyperparameters. Specifically, we adjusted the parameter $\alpha$ in Table 2 to observe its effect on performance. For instance, when $\alpha$ is set to 0.01, the reward of ELEGANT becomes 0.99. Conversely, when $\alpha$ is set to 0.001, the reward of ELEGANT decreases to 0.96.

F.2 DETAILS FOR IMAGE TASKS

Below, we explain the training details and list hyperparameters in Table 7.

F.2.1 FURTHER DETAILS OF IMPLEMENTATION

We use 4 A100 GPUs for all the image tasks. We use the AdamW optimizer (Loshchilov and Hutter, 2019) with $\beta_1 = 0.9, \beta_2 = 0.999$ and weight decay of 0.1. To ensure consistency with previous research, in fine-tuning, we also employ training prompts that are uniformly sampled from 50 common animals (Black et al., 2023; Prabhudesai et al., 2023).

**Sampling.** We use the DDIM sampler with 50 diffusion steps (Song et al., 2020). Since we need to back-propagate the gradient of rewards through both the sampling process producing the latent representation and the VAE decoder used to obtain the image, memory becomes a bottleneck. We employ two designs to alleviate memory usage following Clark et al. (2023); Prabhudesai et al. (2023): (1) Fine-tuning low-rank adapter (LoRA) modules (Hu et al., 2021) instead of tuning the original diffusion weights, and (2) Gradient checkpointing for computing partial derivatives on demand (Gruslys et al., 2016; Chen et al., 2016). The two designs make it possible to back-propagate gradients through all 50 diffusing steps in terms of hardware.

Table 7: Training hyperparameters.

| Hyperparameter | Value |
|---|---|
| Classifier-free guidance weight | 7.5 |
| DDIM steps | 50 |
| Truncated back-propagation step | $K \sim \text{Uniform}(0, 50)$ |
| Learning rate | 0.0001 |
| Batch size | 128 |
| Clip grad norm | 5.0 |

**Guidance.** To train the classifier, we use the AVA dataset (Murray et al., 2012) which includes more than 250k evaluations (i.e., 20 times more samples than our **ELEGANT** implementation, cf. Figure 6). We implement the classifier (i.e., reward model) using an MLP model that takes the concatenation of sinusoidal time embeddings (for time $t$) and CLIP embeddings (Radford et al., 2021) (for $x_t$) as input. The implementation is based on RCGDM (Yuan et al., 2023).

### F.2.2 FURTHER DETAILS OF EVALUATION VIA VISION LANGUAGE MODELS

A key consideration in evaluating all algorithms in the image domain is that we don't know the true $r^\star$. While we use LAION Aesthetic Predictor V2 (Schuhmann, 2022) as $r(x)$, this $r(x)$ is not accurate in out-of-distribution regions, as we mention in the main text. Indeed, when overoptimization happens, generated images become almost identical regardless of prompts.

To effectively detect reward overoptimization, we use a pre-trained multi-modality language model to assess image-to-prompt alignment. For each generated image, we send the following prompt to LLaVA (Liu et al., 2024) along with the image:

```
<image>
USER: Does this image include {prompt}? Answer with Yes or No
ASSISTANT:
```

We assessed its accuracy and precision with human evaluators by generating images using Stable Diffusion with animal prompts (such as dog or cat). The F1 score achieved was 1.0.

### F.2.3 ADDITIONAL RESULTS

**More generated images.** We provide more generated samples to illustrate the performances in Figure 5.

**Comparison with Guidance.** In practice, we observe that the guidance strength in **Guidance** is hard to control: if the guidance level and target level are not strong, the reward-guided generation would be weak (cf. Table 8). However, with a strong guidance signal and a high target value, the generated images become more colorful at the expense of reducing "modified $r$". In presenting qualitative results in Figure 5, we set the target as 10 and the guidance level as 100 to balance guidance strength and "modified $r$".

Table 8: Evaluation results of classifier guidance for aesthetic scores. ($\cdot$) are $95\%$ confidence intervals. Note the top $5\%$ value is 6.0. Modified rewards reflect prompt-image alignments.

| Target ($y_{\text{con}}$) | Guidance level ($\gamma$) | Mean "modified $r$" ↑ | KL-Div ↓ |
|---|---|---|---|
| 6 | 400 | 5.69(0.06) | 0.30 |
| 6 | 800 | 5.71(0.06) | 1.26 |
| 6 | 1200 | 5.68(0.06) | 1.28 |
| 6 | 1600 | 5.45(0.25) | 2.19 |
| 10 | 100 | 5.91(0.14) | 2.52 |
| 10 | 200 | 5.53(0.45) | 6.46 |

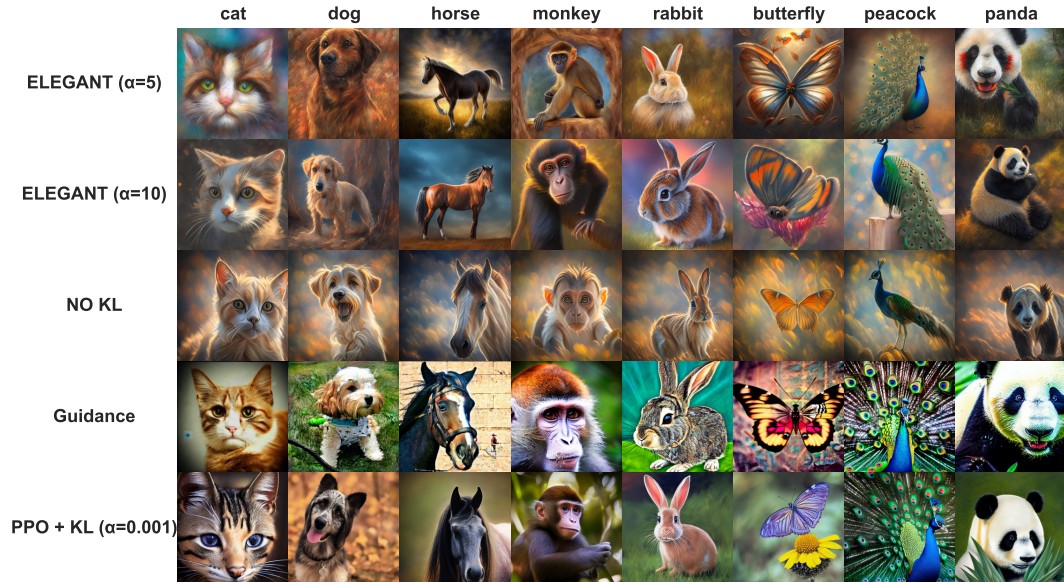

Figure 5: More images generated by **ELEGANT** and baselines. **Guidance** is trained on AVA dataset (Murray et al., 2012). All other algorithms (**NO KL**, **Guidance**, **PPO + KL**, and **ELEGANT**) make 15360 reward inquiries to perform fine-tuning.

**Compared with NO KL, PPO and PPO + KL.** We plot the training curves of **NO KL**, **PPO**, **PPO + KL**, versus **ELEGANT** in Figure 6. Note that this plot depicts a "nominal" reward. Hence, the **NO KL** baseline achieves seemingly high values. However, it severely suffers from overoptimization (See evaluation in Figure 3a and Table 3b). On the other side, it is also evident that the KL entropy of **NO KL** explodes, which indicates that the fine-tuned model deviates from the pre-trained model. Our **ELEGANT** enjoys good performances while keeping a relatively low entropy compared to baselines. This is because our explicit entropy regularization makes balancing fine-tuning and mitigating overoptimization possible.

Empirically, we observe **PPO** outperforms **PPO + KL** in terms of reward but still falls short compared to our **ELEGANT**.

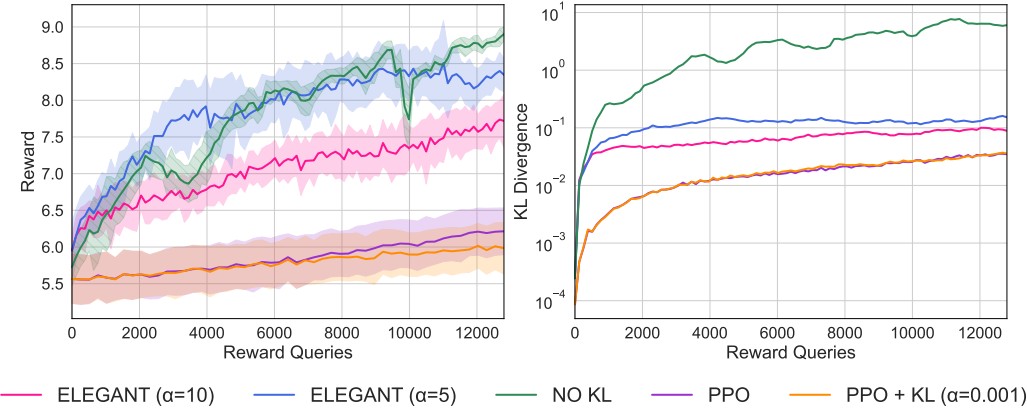

Figure 6: Training curves of reward (left) and KL divergence (right) for **NO KL**, **PPO**, **PPO + KL**, and **ELEGANT** for fine-tuning aesthetic scores. The $x$ axis corresponds to the number of reward queries in the fine-tuning process. All nominal rewards, including baselines, go over 8 in this figure, but baselines still suffer from overoptimization, as in Figure 7. That's why we report with modified instead.

### F.2.4 EFFECTIVENESS OF LLaVA-AIDED EVALUATION

In this section, we see an example of the effectiveness of LLaVA-aided evaluation. More specifically,

Table 9 presents the statistics of LLaVA-aided evaluations for the pre-trained model and 5 checkpoints of the **NO KL** baseline. It is observed that LLaVA can recognize all the prompts of images generated by the pre-trained model. However, even with seemingly high-reward samples, many samples from the **NO KL** ignore their prompts, leading to a decreased "modified $r$".

Table 9: Evaluation statistics of "modified $r$" based on LLaVA

| method | mean | std | max | invalid/total samples |
|---|---|---|---|---|
| pre-trained model | 5.833 | 0.340 | 6.909 | 0/512 |
| NO-KL-ckpt-6 | 7.294 | 0.543 | 7.946 | 2/512 |
| NO-KL-ckpt-7 | 7.379 | 0.796 | 8.139 | 5/512 |
| NO-KL-ckpt-8 | 7.483 | 1.101 | 8.227 | 10/512 |
| NO-KL-ckpt-9 | 6.880 | 2.505 | 8.376 | 59/512 |
| NO-KL-ckpt-10 | 7.025 | 2.612 | 8.730 | 61/512 |

Figure 7 illustrates six instances of failure based on LLaVA evaluation. These examples involve images that disregard prompts, potentially resulting in higher original scores $r$. With our modified $r$, we can adequately assign low scores to such undesired scenarios.

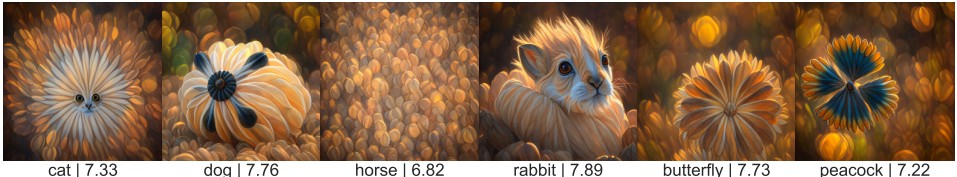

cat | 7.33    dog | 7.76    horse | 6.82    rabbit | 7.89    butterfly | 7.73    peacock | 7.22

Figure 7: Image-prompt alignment failures detected by LLaVA.

## G   Further Limitations and Their Remedy

In this section, we discuss further possible limitations in our work.

### G.1   Computational Cost: Cost of Learning Initial Distributions

Our overhead in learning the initial distribution is low. When learning a second diffusion chain, our goal is to learn $\exp(V_0^\star(x))\nu_{\mathrm{ini}}(x)$. This distribution is much simpler and smoother compared to the target distribution $\exp(r(x)/\alpha)p_{\mathrm{data}}(x)$ in the first diffusion chain. Therefore, we require fewer epochs to learn this distribution. For instance, in image generation, we use only 200 reward queries to learn initial distributions while the main diffusion chain takes 12000 queries for finetuning. The wall time of learning the initial distribution (i.e., the second diffusion chain) is $30 - 40$ minutes in image experiments, while the wall time of learning the second chain is roughly 1800 minutes.

### G.2   Memory Complexity of ELEGANT

When updating a single gradient, while ELEGANT consumes O(L) memory (where L represents the number of discretizations), PPO only requires O(1) memory. This may initially seem like a limitation of our approach. However, in practical scenarios, we are able to manage highly-dimensional data effectively by implementing gradient checkpointing and accumulating gradients while maintaining a large batch size, as we did in our experiments.

### G.3   Choice of $\alpha$

A sophisticated way to choose $\alpha$ is our future work. Typically, we observe $\alpha$ is helpful regardless of its specific choice as long as it is too small or too large enough, as we did ablation studies in Figure 2 and Figure 3c.

Here, we discuss a practical way to choose it and associated experimental results. In many scenarios, we typically know the feasible upper bound of true rewards. In such cases, by appropriately selecting $\alpha$ to ensure that the final learned reward falls within the range of 0.98-0.99 of the upper bound, we can effectively attain high rewards while mitigating overoptimization, as shown in Table 2. Approaches without entropy regularization may easily lead to overoptimization, wherein the learned reward may reach 1, despite the actual reward being relatively low, as we show in Table 2.

### G.4   Inference Time

Inference time is a critical aspect of many works in diffusion models. One potential approach would be to use recent distillation techniques to accelerate inference time after fine-tuning. We will add more such discussion in the next version.

