# OpenReview forum: "Fine-Tuning of Continuous-Time Diffusion Models as Entropy-Regularized Control"
_ICLR.cc/2025/Conference — Submitted to ICLR 2025_

### Official Review · Reviewer_EaxU · 2024-10-30

**Soundness:** 3
**Presentation:** 2
**Contribution:** 3
**Rating:** 5
**Confidence:** 4

**Summary:**

The paper provides a theoretical framework that frames diffusion model fine-tuning as entropy-regularized stochastic optimal control. The ELEGANT method combines terminal reward optimization, entropy regularization against the pre-trained model, and simultaneous learning of both drift terms and initial distributions. This approach maintains the bridges (posterior distributions) of pre-trained models, theoretically proves to stay within the support of the training distribution, and shows connection to classifier guidance. Compared to PPO-based methods, it also demonstrates advantages in computational efficiency.

The researchers evaluated the method on three tasks: GFP (protein) sequence generation, TFBind (DNA) sequence generation, and image generation with aesthetic quality optimization. The experimental results show that ELEGANT outperforms other baseline methods (PPO+KL, Guidance, NO KL) in terms of achieving higher genuine rewards, maintaining lower KL divergence from the pre-trained distribution, and training speed and efficiency.

**Strengths:**

The originality of this work is out of question. This paper address the diffusion alignment problem via the continuous-time stochastic control perspective. The quality and the clarity of this paper is reasonable. The authors have conducted a series of numerical experiments to demonstrate its effectiveness.

Advantages:
- Maintains bridges (posterior distributions) of pre-trained models
- Theoretically proven to stay within support of training distribution
- Shows connection to classifier guidance
- More computationally efficient than PPO-based methods
- Better at mitigating overoptimization

**Weaknesses:**

Algorithm 1 presents a method with three stages: value function learning and solving two neural SDEs. This seems computational heavy, especially the Neural SDE solving should be hard due to the storation of computation graph when the SDE is parameterized by huge neural networks. It seems impossible for stable diffusion as usually it is only possible to forward pass with batch size = 4 on 80G GPU, let alone the computation graph along a trajectory. I wonder if more clarification can be made about the computation implementation and speed.

There are several other works that should be compared or mentioned:
- Adjoint Matching: Fine-tuning Flow and Diffusion Generative Models with Memoryless Stochastic Optimal Control
- Improving GFlowNets for Text-to-Image Diffusion Alignment, 2024
- Implicit diffusion: Efficient optimization through stochastic sampling, 2024.

The pretrained diffusion model is not continuous time. How to adapt it to the framework proposed in this paper?

There is severe reward overoptimization issue for aesthetic score. Would there be a trade-off when we tune the value of alpha?

The writing is not clear in some places. For example, is f, u, q parameterized neural networks, or abstract notation? Does v mean value function or initial distribution? I can roughly guess their meaning, but this may not be easy for people who are not familiar with SDE.

**Questions:**

N/A

---

> ### Author Response · Authors · 2024-11-18
>
> We thank the reviewer for their feedback. We have addressed your concerns by providing explanations on (1) the implementation with Stable Diffusion, (2) additional discussion of related work, and (3) the choice of $\alpha$. We are happy to answer more if further clarification is needed.
>
> **Q. Algorithm 1 presents a method with three stages: value function learning and solving two neural SDEs. This seems computational heavy, especially the Neural SDE solving should be hard due to the storation of computation graph when the SDE is parameterized by huge neural networks. It seems impossible for stable diffusion as usually it is only possible to forward pass with batch size = 4 on 80G GPU, let alone the computation graph along a trajectory. I wonder if more clarification can be made about the computation implementation and speed.**
>
> A. Thank you for pointing this out. **We utilized standard memory-efficient fine-tuning techniques for diffusion models, as described in [1] and [2]. Hence, memory efficiency is not a substantial practical bottleneck in our experiments.** While the main paper does not go into these details due to space limitations, we have provided an extensive discussion of the implementation details and computational costs in Appendix F.2.1 and Appendix G. Below is a summary:
>
> 1. As outlined in Appendix F.2.1, we employed two strategies to reduce the memory footprint of direct reward backpropagation, following [1,2]:
> * Fine-tuning only the LoRA modules rather than the original diffusion model weights.
> * Utilizing gradient checkpointing to compute derivatives on the fly.
> These techniques enable backpropagation through all 50 diffusion steps of Stable Diffusion (using a DDIM scheduler) within hardware constraints. Specifically, leveraging these methods allows fine-tuning Stable Diffusion with a batch size of 4 to consume approximately 30 GB of GPU memory.
> * Additionally, we use gradient accumulation and multiple GPUs to scale the effective batch size to 128, facilitating smoother optimization.
>
> 2. The two neural SDEs are optimized separately and sequentially. Once the initial distribution is obtained, the optimization of the second neural SDE does not require storing computational graphs for parameters from the first SDE.
>
> [1] Clark, Kevin, et al. "Directly fine-tuning diffusion models on differentiable rewards." ICLR 2024
>
> [2] Prabhudesai, Mihir, et al. "Aligning text-to-image diffusion models with reward backpropagation." ICLR 2024
>
> **Q. The pretrained diffusion model is not continuous time. How to adapt it to the framework proposed in this paper?**
>
> A. Regarding the distinction between continuous-time and discrete-time training in diffusion models, we interpret the reviewer’s comment as referring to the former as training with randomly sampled time steps and the latter as training with fixed time steps. When pre-trained models are trained in a discrete-time training way, in solving the Neural SDE within our approach, the most natural strategy is to adopt the same discretization scheme as used in the pre-trained model, which is precisely the approach taken in our algorithm.
>
> **Q. There are several other works that should be compared or mentioned**
>
> A. Thank you for your suggestion. We now acknowledge that all of the referenced works are related to ours. We have incorporated it into a new version of our draft.
>
> *[1] and [2] have addressed problems similar to ours, but their approaches still differ significantly. [1] addressed the initial bias issue discussed in Section 6 by modifying the noise schedule, whereas we tackled it by introducing an additional optimization problem. [2] approached a related problem by designing an objective function inspired by a detailed balance loss in Gflownets, while our algorithm directly solves the control problem using neural SDE. While their algorithm has certain benefits when rewards are differentiable like PPO, our primary focus is more on how to mitigate overoptimization from both theoretical and empirical perspectives. [3] has explored fine-tuning in diffusion models, framing it as a bi-level optimization problem. However, they do not appear to discuss strategies for constructing objective functions, such as incorporating KL regularization to prevent over-optimization.*

---

> > ### Author Response · Authors · 2024-11-18
> > **(Continue)**
> >
> > **Q. There is severe reward overoptimization issue for aesthetic score. Would there be a trade-off when we tune the value of alpha?**
> >
> > We have shown that our algorithms can effectively mitigate reward overoptimization in Section 8.3 using a carefully designed metric. While the main paper does not go into these details of the choice of $\alpha$ due to space limitations, we have actually discussed this point in Appendix G.3. as follows. Note that Appendix F.2.4 also discuss how we evaluate overoptimization with aesthetic scores more.
> >
> > *Choice of $\alpha$: A sophisticated way to choose $\alpha$ is our future work. Typically, we observe $\alpha$ is helpful regardless of its specific choice as long as it is too small or too large enough, as we did ablation studies in Figire 2 and Table 3. Here, we discuss a practical way to choose it and associated experimental results.  In many scenarios, we typically know the feasible upper bound of true rewards. In such cases, by appropriately selecting $\alpha$ to ensure that the final learned reward falls within the range of $0.98$-$0.99$ of the upper bound, we can effectively attain high rewards while mitigating overoptimization, as shown in Table 3. Approaches without entropy regularization may easily lead to overoptimization, wherein the learned reward may reach $1$, despite the actual reward being relatively low, as we show in Table 3.*
> >
> > **Q. The writing is not clear in some places. For example, is f, u, q parameterized neural networks, or abstract notation? Does v mean value function or initial distribution? I can roughly guess their meaning, but this may not be easy for people who are not familiar with SDE.**
> >
> > We use abstract notation; however, since neural networks are employed in practice, we have added this point further in a new version. Regarding $v$ and $\nu$, we use the Greek letter $\nu$ for the initial distribution, which may resemble the notation for value functions $v$. We use a capital letter in a new version.

---

> > > ### Author Response · Authors · 2024-11-30
> > > **From Authors**
> > >
> > > Thank you for your thoughtful feedback. We sincerely appreciate the time and effort you dedicated to reviewing our submission. Could you kindly check our rebuttal? We have updated our draft to reflect your comments.

---

### Official Review · Reviewer_VVwy · 2024-11-01

**Soundness:** 4
**Presentation:** 2
**Contribution:** 2
**Rating:** 6
**Confidence:** 3

**Summary:**

The paper presents a novel approach for fine-tuning diffusion models. The proposed method, ELEGANT, aims to address the challenge of overoptimization when fine-tuning diffusion models based on a learned reward function. Overoptimization occurs when the model produces samples with high nominal rewards that are outside the support of the data distribution on which the diffusion model was trained. To prevent overoptimization, the authors propose an entropy-regularized objective that is optimized over trajectories, i.e., path measures, formulated as a stochastic optimal control problem.  The authors introduce another learnable SDE to obtain approximate samples from the intractable initial distribution. The numerical evaluation shows that the proposed approach is on par or better compared to the considered baselines in terms of reward while achieving a lower KL divergence.

**Strengths:**

- The problem of overoptimization is well-motivated and clearly explained.
- Using entropy-regularization is a sensible and approach to counteract overoptimization.
- Interesting connections and explanations, such as Feynman-Kac, Bridge Preservance, and Classifier Guidance.
- Numerical results show that entropy regularization helps to stay closer to the support of the original diffusion model.

**Weaknesses:**

Things that might improve the clarity and readability of the paper:

- Readers unfamiliar with KL divergences on path measures might wonder where the objective function (5) is coming from. I think it would be great to include a derivation (which is already part of the proof of Theorem 1 but it would be good to make it more explicit).
- The same holds for the optimal entropy-regularized value function (Equation after Eq. (5)).
- The authors make it sound like the Feynmac-Kac (FK) formulation is introduced to provide intuition. However, to the best of my knowledge, it is used to learn the approximate optimal initial value function $\hat a$. It would be helpful to make this connection more explicit in the main part of the paper.
- Moreover, the optimal initial value function $v_0^*$ is used in objective Eq. (8). Only later in the paper is explained how it is obtained. The authors should at least include a sentence that an explanation for how to obtain $v_0^*$ is deferred to later in the document.
- The text between Lemma 1 and 2 uses an expression for the optimal drift that, to the best of my knowledge, stems from the HJB equation. This is never mentioned in the main text and potentially could cause confusion. It would be good to explain where the formula for the optimal drift is coming from
- From my perspective, the main body of the paper contains content that is not integral to understanding the methodology, such as the section on Diffusion Bridges or Algorithm 2 & 3 and instead defers parts that are essential for understanding the paper to the appendix, in particular, the approximation of optimal initial value function.
- The term entropy regularization actually refers to a KL regularization which could potentially cause confusion.
- Using a capital letter for describing the value function may help to distinguish it from the initial distribution $\nu$ due to their similarity.

Weaknesses:

- While inference time is already a bottleneck for diffusion models, the proposed approach requires simulating two SDEs.
- The approach requires solving two stochastic optimal control problems and an additional regression problem for obtaining the initial value function.
- The method requires a) gradient checkpointing and b) low-rank modules to train, making the approach inconvenient to use.
- My main concern with the paper is the following: The terminal reward $v_0^*$ for learning the drift $q$ is approximated with a neural network using a regression objective. Hence, the learned model could be arbitrarily bad in regions with no data which is also the cause for the overoptimization problem. It thus seems like the problem has just shifted to the initial value function estimation and thus not properly addressed.

**Questions:**

See Weaknesses.

---

> ### Author Response · Authors · 2024-11-17
> **Re:**
>
> We sincerely appreciate your detailed and thoughtful feedback. In this response, we have addressed your primary concerns regarding our algorithms. We are happy to answer more if further clarification is required. Regarding your suggestions for the presentation, we have incorporated many of them, and the change is in red. (For some of the suggestions, like replacing sections in the main text and appendix, we plan to do; but at this moment, to avoid confusion for other reviewers, we didn't implement it)
>
> **Weakness: My main concern with the paper is the following: The terminal reward $v_0$ for learning the drift $q$ is approximated with a neural network using a regression objective. Hence, the learned model could be arbitrarily bad in regions with no data which is also the cause for the overoptimization problem. It thus seems like the problem has just shifted to the initial value function estimation and thus not properly addressed.**
>
> While we acknowledge your observation, we think it won't be a primary concern for the following reasons.  In summary, we do not encounter the distribution shift issue in this context because error guarantees are required solely within the original data distribution. Specifically, the reasoning is as follows:
>
> First, recall from Corollary 2, the optimal initial distribution is
> $$
>  \exp(v^{ \star}\_0( \cdot)/ \alpha) \nu\_{ini}(\cdot)/C.
> $$
> Then, leveraging the relation from Lemma 1:
> \begin{align*}
>      \exp\left(\frac{v^{\star}\_{0}(x)}{\alpha} \right)= \mathrm{E}\_{\mathbb{P}^{data}}\left [\exp \left (\frac{r(x_T)}{\alpha } \right)|x_0=x \right],
> \end{align*}
> our algorithm approximates the value function $v^{\star}$ in the optimal initial distribution with supervised learning:
> \begin{align*}
>      \hat g = \mathrm{argmin}\_{g \in \mathcal{G}} \sum\_{i=1}^n  \left ( \exp   \left (\frac{r(x^i_T)}{\alpha } \right )- g(x^i_0) \right )^2,  x^i_0 \sim \nu\_{ini}, x^i_T\sim \mathbb{P}^{data}(x_0),
> \end{align*}
> where $\mathcal{G}$ denotes a function class, such as neural networks. Under the well-specification assumption (a standard assumption as used in Wainright 2019), we achieve the following mean squared error (MSE) guarantee:
> \begin{align*}
>         \mathrm{E}\_{x_0\sim \nu\_{ini}}\left [ \left (\hat g(x_0)- g^{\star}(x_0) \right)^2 \right ] = O\left (\frac{Cap(\mathcal{G})}{n} \right)
> \end{align*}
> where $Cap(\mathcal{G})$ represents the capacity of the function class $\mathcal{G}$ and $g^{\star}$ is $\exp(v_0(\cdot)/\alpha)$.
>
> This result implies that we obtain the MSE guarantee within the original initial distribution $\nu\_{ini}$. Although this guarantee does not extend beyond the support of $\nu\_{ini}$, it is not an issue because we know that the support of the optimal initial distribution is restricted to $\nu\_{ini}$ (recalling Corollary 2), and in the second step of approximating the optimal initial distribution, in our algorithm, we consistently do sampling on the support of the original initial distribution $\nu\_{ini}$.
>
> **Q. While inference time is already a bottleneck for diffusion models, the proposed approach requires simulating two SDEs.The approach requires solving two stochastic optimal control problems and an additional regression problem for obtaining the initial value function.**
>
> While we ackowldge your opinion, we have an explanation that this won't be a concern in Appendix G.1. In particualr, the following paragraph demonstrates that learning the optimal initial distribution is not a practical bottleneck for our algorithm.
>
> *Computational Cost: Cost of Learning Initial Distributions: Our overhead in learning the initial distribution is low. When learning a second diffusion chain, our goal is to learn $\exp(v^{\star}_0(x))\nu\_{\mathrm{ini}}(x)$. This distribution is much simpler and smoother compared to the target distribution $\exp(r(x)/\alpha)p\_{\mathrm{data}}(x)$ in the first diffusion chain. Therefore, we require fewer epochs to learn this distribution. For instance, in image generation, we use only $200$ reward queries to learn initial distributions while the main diffusion chain takes $12000$ queries for finetuning. The wall time of learning the initial distribution (i.e., the second diffusion chain) is $30-40$ minutes in image experiments, while the wall time of learning the second chain is roughly $1800$ minutes.*
>
> **Weakness. The method requires a) gradient checkpointing and b) low-rank modules to train**
>
> This technique is commonly employed when fine-tuning large models. Representative works for fine-tuning diffusion models (Black et al., 2023; Prabhudesai et al., 2023; Clark et al., 2023) also utilize these techniques, indicating that this is not a unique aspect of our work. Hence, we believe this is not a particular weakness of our methods. Additionally, we clarify that for smaller models, such as those used in experiments with biological diffusion models, employing this technique is not necessary.

---

> > ### Comment · Reviewer_VVwy · 2024-11-25
> > **Re:**
> >
> > We thank the authors for their reply.
> >
> > ---
> > > While we acknowledge your observation, we think it won't be a primary concern for the following reasons.
> >
> > I apologize for not understanding the author's explanation. Let me rephrase my concern:
> > How is learning the drift $\hat{q}$ for the model that generates the initial distribution prevented from putting probability mass on support where the optimal initial value function $\hat{a}$ was not trained? From my understanding, the following optimization problem
> > \begin{equation}
> > \hat{q} = \text{argmax}_q \mathbb{E}_q \left[\hat{a}(x_0) -  \frac{\alpha}{2} \int \frac{\|q\|}{\tilde{\sigma^2}} dt \right]
> > \end{equation}
> > does not prevent $\hat{q}$ from putting probability mass on support where the optimal initial value function $\hat{a}$ is not trained.

---

> ### Author Response · Authors · 2024-11-25
>
> Thank you for your response and engagement. Below, we provide additional clarification regarding your concerns. We are happy to clarify more if specific points remain unclear.
>
> 1. First, we would like to emphasize the assumptions outlined in Section 6:
>
> *We start with a reference SDE over the interval $t \in [-T,0]; t \in [-T,0];   d x_t = \tilde \sigma(t)dw_t, x_{-T}=x_{fix},$
> such that the distribution at time $0$ follows $\nu_{ini}$.*
>
> 2. Under the above assumption, we have concluded in Theorem 2 that the generated distribution by
> \begin{align*}
>   t \in[-T,0];  d x_t  = q^{\star}(t, x_t)dt + \tilde \sigma(t) dw_t
> \end{align*}
> is $$\hat a(x_0)\nu_{ini}(x_0)/C,$$ where $q^{\star}$ is defined by the following control problem
> \begin{align}
> q^{\star}=\arg \max_{q}\mathrm{E}\_{P^{q}} \left[\hat a(x_0)-0.5 \alpha \int^0_{-T} \frac{\|q(t,x_t)\|^2}{\tilde \sigma^2(t) }dt  \right], \quad  (a)
> \end{align}
> Our control problem (a) incorporates the KL divergence as the second term (the integral term), which ensures that the optimization remains within the support of $\nu_{ini}$. To provide a high-level explanation of why the support is preserved, , consider a simplified problem below.
>
> *  For an optimization problem where $q$ is any distribution over $X$, $\nu_{ini}$ is a fixed distribution, and $\hat a(\cdot)$ represents a value function:
> \begin{align*}
> q^{\star} = \arg\max_{q \in \Delta(X)} E_{x \sim q}[\hat a(x)] - \mathrm{KL}(q|| \nu_{ini})
> \end{align*}
> In this problem, we can easily show the optimal distribution $\forall x \in X, q^{\star}(x) = \hat a(x)\nu_{ini}(x)/C$ [1]. Thus, $q^*$ will retain the support of $\nu_{init}$ over $X$.
>
> Returning to our control problem (a), the second term (i.e., the integral term) corresponds to the KL divergence between the path measures on $x_{0:T}$ induced by two SDEs—one that needs to be learned and another corresponding to $\nu_{ini}$ (formally shown using Girsanov theorem [2] in our proof). Consequently, this term similarly ensures that the distribution induced by the fine-tuned SDE remains within the distribution of $\nu_{ini}$.
>
>
> * At this stage, note this theorem is unrelated to the function approximation of value functions ($\hat a(x_0)$)
>
> 3. Now, let me connect the above discussion with the function approximation error for $\hat a$. As discussed in our first rebuttal, by invoking the standard guarantees of regression, we obtain the mean squared error (MSE) guarantee for $\hat a(x_0)$ under the initial distribution:
> \begin{align*}
>     \mathrm{E}\_{x_0 \sim \nu_{ini}}[ ( \hat a(x_0)- a(x_0) )^2].
> \end{align*}
> Thus, there is no distributional shift, as the MSE guarantee for $\hat a$ is explicitly on the support of $\nu_{ini}$ and we are sampling from $\hat a(x_0)\nu_{ini}(x_0)$, as previously stated in the second step.
>
>
> [1] Ziegler, Daniel M., et al. "Fine-tuning language models from human preferences." arXiv preprint arXiv:1909.08593 (2019).
>
> [2] Shreve, Steven E. Stochastic calculus for finance II: Continuous-time models. Vol. 11. New York: springer, 2004.

---

> > ### Comment · Reviewer_VVwy · 2024-11-26
> > **Reply to authors**
> >
> > I thank the authors for the details. My concern has been resolved. I will raise my score to 6.

---

> > > ### Author Response · Authors · 2024-11-27
> > >
> > > Thanks for reading our response and adjusting the score! We are happy that our clarifications helped address your concerns and are thankful for your constructive comments.

---

### Official Review · Reviewer_rDSc · 2024-11-03

**Soundness:** 3
**Presentation:** 3
**Contribution:** 2
**Rating:** 6
**Confidence:** 3

**Summary:**

The authors propose a new method for fine-tuning diffusion models by maximizing the value of the reward functioning in a goal directed manner. They propose an objective that maximizes a reward function towards the desired goal, jointly with the negative KLD to the data distribution (to stay close to the pre-trained diffusion model). The closed-form expression of the optimal solution is an energy based model (untractable). The authors show that they can augment the pre-trained diffusion model to sample from this EBM. This is achieved by adding a drift term and learning the initial distribution.

**Strengths:**

* The problem is interesting. The approach of augmenting a diffusion model with a term to approximate a different distribution is elegant.

* Approximating EBMS is a hard problem. This seems to be another nice solution in another useful special setup.

**Weaknesses:**

* In the write-up, the authors state that the methods is an entropy-regularized control against the pretrained diffusion model but in the equations, the data distribution is used instead of the pre-trained distribution. Could you clarify?

* Approximation errors when solving of E9 and E10. Can [1] be used instead? Similarly to the proposed approach, the entropy of the EBM policy can be approximated in closed form. The parameters of the samplers (also initial distribution) can be simply learnt via optimizing the KLD. [1] seems to be more parameter and computationally efficient.

[1] Messaoud, Safa, et al. "S $^ 2$ AC: Energy-Based Reinforcement Learning with Stein Soft Actor Critic." arXiv preprint arXiv:2405.00987 (2024).

* Missing references to proofs in appendix (eq5, theorem 1, corolalaries 1 and 2)

**Questions:**

(1) Can p_{data} be replace by the distribution of the pre-trained model? What are the pros and cons of using each of these distributions.

(2) Why is your approach of modifying the diffusion process better than using a parametrized SVGD as in [1]?

[1] Messaoud, Safa, et al. "S $^ 2$ AC: Energy-Based Reinforcement Learning with Stein Soft Actor Critic." arXiv preprint arXiv:2405.00987 (2024).

---

> ### Author Response · Authors · 2024-11-19
>
> We appreciate the reviewer’s positive feedback.
>
> **Q. In the write-up, the authors state that the methods is an entropy-regularized control against the pretrained diffusion model but in the equations, the data distribution is used instead of the pre-trained distribution. Could you clarify?**
>
> Your understanding is correct. We have considered a scenario where the pre-trained distribution closely approximates the data distribution, assuming the availability of a foundational pre-trained generative model trained on a large dataset of natural samples. Meanwhile, the data with feedback remains significantly more limited. This setting is practical as we mention in the introduction. We have clarified this assumption more explicitly in the updated version of the manuscript.
>
> **Approximation errors when solving of E9 and E10. Can [1] be used instead?**
>
> Thanks for the suggestion. This approach could potentially be applied. However, in practice, we observe that explicitly approximating Q-functions (e.g., critic models) is challenging in the context of fine-tuning diffusion models due to the high dimensionality of the state space. Therefore, we employ purely policy-based optimization methods to solve our optimization problem. While incorporating approaches such as SAC, SQL, or S^2AC in [1] is indeed an intriguing direction, we leave this exploration for future work.

---

### Official Review · Reviewer_KWus · 2024-11-03

**Soundness:** 3
**Presentation:** 3
**Contribution:** 3
**Rating:** 6
**Confidence:** 3

**Summary:**

This study focuses on the overoptimization issue in the finetuning of diffusion models. The authors point out that recent methods of finetuning diffusion models usually suffer from the overoptimization problem, which shifts the generation distribution towards a high nominal reward value but away from the natural distribution. To this end, the authors propose a new finetuning method to adjust the generation distribution. They reformulate SDE with an additional drift term and a different initial distribution. Then, they provide the optimal solution to the drift term and initial distribution and develop an optimization algorithm to solve it. Experimental results on biological sequence tasks and image generation demonstrate that the proposed method achieves high reward values while keeping a small value of KL divergence from the original distribution.

**Strengths:**

+ The overoptimization problem is important in finetuning diffusion models, and this study provides a sound way to address it. The authors formally formulate the overoptimization problem and the desired properties in Section 4.
+ This study involves both solid theoretical analysis and interpretations that are easily understood.
+ The evaluation based on the comparison between nominal reward and genuine reward is interesting and persuasive.

**Weaknesses:**

- The motivation for learning a different initial distribution is unclear. Why does the initial distribution affect the finetuning of diffusion models, and why should we finetune the model on a different initial distribution? On the other hand, does it affect the model performance on general tasks? In other words, given the model finetuned with the learned initial distribution, if users sample the generation from the original Gaussian distribution in the sampling phase, how about the quality of generations?
- Equations in Lines 372-377 are slightly confusing. What do $y_0$, $y_t$, and $y_T$ mean and how to obtain them in implementation?
- The connection between the added drift term and the classifier guidance is interesting. Given that they have a similar mathematical formulation, I expect a theoretical analysis of their difference in performance.
- How was $\alpha$ in PPO+KL set in experiments of Figure 3? Have you tried different settings of $\alpha$ for PPO+KL and ELEGANT for comparison? I notice that in Figure 5 and Figure 6, $\alpha$ was set to 0.001 for PPO+KL, which was very small, while ELEGANT used $\alpha=\{5,10\}$. I am not sure whether such a comparison is fair.
- Generations of ELEGANT still exhibit the overoptimization phenomenon. For example, the first line of Figure 3(c), especially the image of ``panda,’’ looks unnatural. Images in Figure 5 also have unnatural backgrounds and colors.
- What is the additional computation cost and memory cost of learning $q$ and $u$?

**Questions:**

Please refer to the above weakness part.

---

> ### Author Response · Authors · 2024-11-23
>
> We appreciate your positive feedback. We have addressed your concerns by providing (1) an additional explanation regarding the significance of learning initial distributions, (2) a clarification of the choice of $\alpha$, and (3) an explanation of how the generated images mitigate overoptimization.
>
> **Q. The motivation for learning a different initial distribution is unclear. Why does the initial distribution affect the finetuning of diffusion models, and why should we finetune the model on a different initial distribution? On the other hand, does it affect the model performance on general tasks? In other words, given the model finetuned with the learned initial distribution, if users sample the generation from the original Gaussian distribution in the sampling phase, how about the quality of generations?**
>
> Thank you for raising this point. Intuitively, in many diffusion models, the initial point could retain significant information about the entire trajectory, making its optimization beneficial during fine-tuning. As an extreme case, in specific diffusion models like rectified flow [1], the initial points contain all trajectory information, as trajectories do not overlap.  Here is a more concrete discussion. We have added some of them in the main text.
>
> * Empirical viewpoint:  **In Section 8.4, we compare the performance of our method with that of original initial distributions (Gaussian distributions)**. Our results demonstrate that incorporating initial distribution learning enhances performance. Notably, two independent studies [2,3] also validate the effectiveness of learning initial distributions (but in different ways from ours).
>
> * Theoretical viewpoint:  Theorem 1 demonstrates that learning an initial distribution enables sampling from the target distribution $p(x)\exp(r(x))$. However, we do not intend to cliam that good practical performance is unattainable without learning an initial distribution. Empirically, we observe that learning an initial distribution enhances performance, but certain good performance can still be achieved without it, as evidenced in Section 8.4.
>
> [1] Liu, Xingchao, Chengyue Gong, and Qiang Liu. "Flow straight and fast: Learning to generate and transfer data with rectified flow." arXiv preprint arXiv:2209.03003 (2022).
>
> [2] Domingo-Enrich, Carles, et al. "Adjoint matching: Fine-tuning flow and diffusion generative models with memoryless stochastic optimal control." arXiv preprint arXiv:2409.08861 (2024).
>
> [3] Ben-Hamu, Heli, et al. "D-Flow: Differentiating through Flows for Controlled Generation." arXiv preprint arXiv:2402.14017 (2024).
> Equations in Lines 372-377 are slightly confusing.
>
> **Q. Equations in Lines 372-377 are slightly confusing.**
>
> A. Thank you for catching it. We have provided additional explanations, marked in red.
>
> **Q. The connection between the added drift term and the classifier guidance is interesting. Given that they have a similar mathematical formulation, I expect a theoretical analysis of their difference in performance.**
>
> A. Thank you for your encouraging feedback. In general, classifier guidance requires differentiable value function models, which necessitates realizability within the value function class. Conversely, our fine-tuning algorithm relies on realizability within the policy class.  We are leaving a more detailed analysis as future work.
>
> **Q. How was α  in PPO+KL set in experiments of Figure 3?**
>
> A. Thank you for pointing this out. We experimented with several values of $\alpha$ (like $\alpha=1,0.1, 0.01$). For higher value of $\alpha$, we observed that the PPO learning curve slows significantly, making it difficult to optimize rewards. Consequently, we have included only the best result. We will provide a more detailed clarification in the next version.

---

> > ### Author Response · Authors · 2024-11-23
> > **(Continue)**
> >
> > **Q. Generations of ELEGANT still exhibit the overoptimization phenomenon. For example, the first line of Figure 3(c), especially the image of ``panda,’’ looks unnatural. Images in Figure 5 also have unnatural backgrounds and colors.**
> >
> > We acknowledge the perspective raised, but we would like to convey that overoptimization is effectively mitigated in our image experiments.
> >
> > * Our goal is to generate images beyond “natural images”, i.e.,  aesthetic images that maintain a degree of naturalness consistent with the prompts. Given that aesthetic images could be abstract or potentially bit unnatural images, this result is expected. Notably, similar images are still treated as aesthetic in prior works (e.g., Section 5.1. In Clark et al., 2023).
> >
> > * **When overoptimization occurs, the generated images deviate significantly from the prompts. This case is demonstrated in Figure 7 of our appendix.** In contrast, the general images in Figures 3 and 5 from our algorithm align closely with the prompts, as confirmed both qualitatively and quantitatively. The LLava evaluation in Section 8.3 further substantiates this alignment. Furthermore, upon visual inspection, it is observed that the images adhere to the intended prompts.
> >
> > **Q. What is the additional computation cost and memory cost of learning q and u?**
> >
> > A. We acknowledge your point. The computational cost is discussed in Appendix G.1.
> >
> > *Computational Cost: Cost of Learning Initial Distributions: Our overhead in learning the initial distribution is low. When learning a second diffusion chain, our goal is to learn $\exp(v^{\star}_0(x))\nu\_{\mathrm{ini}}(x)$. This distribution is much simpler and smoother compared to the target distribution $\exp(r(x)/\alpha)p\_{\mathrm{data}}(x)$ in the first diffusion chain. Therefore, we require fewer epochs to learn this distribution. For instance, in image generation, we use only $200$ reward queries to learn initial distributions while the main diffusion chain takes $12000$ queries for finetuning. The wall time of learning the initial distribution (i.e., the second diffusion chain) is $30-40$ minutes in image experiments, while the wall time of learning the second chain is roughly $1800$ minutes.*
> >
> > The above does not address the memory issue, but we use a much smaller model for initial distribution learning. Hence, memory won’t be a practical issue.

---

### Meta-Review · Area_Chair_GWmA · 2024-12-19

**Metareview:**

The paper considers the problem of fine-tuning of diffusion models for the given reward function. Namely, it proposes to maximize given reward function regularized with an additional KL penalty term, which amounts to sampling from the following product of densities
$$p_{\text{target}} \propto p_{\text{pretrained}}(x)\exp(\beta r(x)),$$
where $p_{\text{pretrained}}(x)$ is the density of samples produced by the pre-trained diffusion model, $r(x)$ is a predefined reward model, and $\beta$ is the inverse temperature hyperparameter.

To solve this sampling problem, the authors propose to learn the initial distribution and the additional drift term to guide the pretrained diffusion dynamics. Their experiments for image generation demonstrate higher reward values while maintaining low KL divergence with the samples from the pre-trained model.

Overall, none of the reviewers is ready to champion the acceptance of this paper, which is reflected in "marginal acceptance" scores and possible uncertainties of the reviews (their confidence). One of the potential improvements of the paper is its presentation, which was extensively commented on by Reviewer VVwy and reflected in the scores of the reviewers EaxU and VVwy. Another issue to address is the computational complexity of the proposed solution, which EaxU and VVwy have noted.

**Additional Comments On Reviewer Discussion:**

- Reviewer VVwy provided extensive feedback on the readability and presentation of the paper.
- Reviewers rDSc, and EaxU proposed additional literature to compare against. The authors have argued that these papers are concurrent works, which I noted and did not include in the evaluation of the paper.
- Reviewer VVwy had conceptual concerns regarding the feasibility of the proposed solution, which, however, were successfully resolved by the authors.

---

### Decision · Program_Chairs · 2025-01-22

Reject